# Stochastic search and joint fine-mapping increases accuracy and identifies previously unreported associations in immune-mediated diseases

Jennifer L. Asimit [1], Daniel B. Rainbow [2], Mary D. Fortune [1], Nastasiya F. Grinberg [3], Linda S. Wicker [2] & Chris Wallace [1,3]

Thousands of genetic variants are associated with human disease risk, but linkage disequilibrium (LD) hinders fine-mapping the causal variants. Both lack of power, and joint tagging of two or more distinct causal variants by a single non-causal SNP, lead to inaccuracies in fine-mapping, with stochastic search more robust than stepwise. We develop a computationally efficient multinomial fine-mapping (MFM) approach that borrows information between diseases in a Bayesian framework. We show that MFM has greater accuracy than single disease analysis when shared causal variants exist, and negligible loss of precision otherwise. MFM analysis of six immune-mediated diseases reveals causal variants undetected in individual disease analysis, including in *IL2RA* where we confirm functional effects of multiple causal variants using allele-specific expression in sorted CD4$^+$ T cells from genotype-selected individuals. MFM has the potential to increase fine-mapping resolution in related diseases enabling the identification of associated cellular and molecular phenotypes.

[1] MRC Biostatistics Unit, University of Cambridge, Cambridge Institute of Public Health, Forvie Site, Robinson Way, Cambridge Biomedical Campus, Cambridge CB2 0SR, UK. [2] JDRF/Wellcome Diabetes and Inflammation Laboratory, Wellcome Trust Center for Human Genetics, Nuffield Department of Medicine, University of Oxford, Oxford OX3 7BN, UK. [3] Department of Medicine, Cambridge Biomedical Campus, University of Cambridge, Box 157, Level 4, Cambridge CB2 0QQ, UK. Correspondence and requests for materials should be addressed to J.L.A. (email: jennifer.asimit@mrc-bsu.cam.ac.uk) or to C.W. (email: cew54@cam.ac.uk)

The underlying genetic contribution to many complex diseases and traits has been investigated with great success by genome-wide association studies (GWAS). Various approaches have identified thousands of variants associated with a spectrum of diseases. In particular, much progress has been made in the genetics of immune-mediated diseases (IMD), revealing a complex pattern of shared and overlapping genetic etiology[1,2].

Fine-mapping—the process of distinguishing causal genetic variants from their neighbours—is an essential step to enable the design of functional assays required to understand the mechanism by which the region impacts disease risk, but it is complicated by linkage disequilibrium (LD)[3]. The problem is often approached through stepwise regression[4,5], which assumes that statistical inference of the best joint model (i.e. a model with multiple causal SNPs) can be derived by starting with the most significant SNP, then conditioning on this and adding the next most significant, continuing this conditioning until no conditionally significant SNPs remain. It has been noted that the SNP with the smallest p-value need not be causal, especially if it is in LD with two causal SNPs[6]. Alternative Bayesian fine-mapping methods have been developed, which use a stochastic search instead of stepwise search[7–9]. Stepwise and stochastic search results may disagree[9] and although stochastic search generally demonstrates improved accuracy[10] these techniques have not yet been widely adopted.

Here, we systematically compare stepwise and stochastic approaches by application to dense genotype data for six IMD, aiming to address the frequency and causes of disagreement between results. We find that stochastic search solutions are more likely to be correct than stepwise search results when sample sizes are large, but that they can face similar issues to stepwise searches when sample sizes are small. We also observe a striking sharing of causal variants between different IMD, consistent with previous reports[1,2], which motivates us to propose a Bayesian multinomial stochastic search method, in which multiple related diseases can be simultaneously fine-mapped. This allows us to borrow information between diseases and achieve correct fine-mapping solutions at smaller sample sizes than when considering individual diseases alone. We show that posterior probabilities under our proposed model can be decomposed into quantities available from single disease analyses, allowing it to be applied without excessive additional computational overhead.

## Results

**Stochastic and stepwise search differences in 10% of regions.** We systematically applied stepwise and stochastic search fine-mapping to dense genotyping data from ImmunoChip studies of six IMD: type 1 diabetes (T1D)[11], multiple sclerosis (MS)[12], autoimmune thyroid disease (ATD)[13], celiac disease (CEL)[14], juvenile idiopathic arthritis (JIA)[15] and rheumatoid arthritis (RA)[16] (sample sizes given in Supplementary Table 1) in 90 densely mapped regions with at least one associated disease (Supplementary Data 1), 204 disease-region combinations in total. Results are given in Supplementary Data 2–3. For RA and CEL, we performed parallel analyses in UK-only and UK + international samples (iRA and iCEL, respectively).

Unlike stepwise search which produces a single best model, stochastic search results are a posterior probability distribution across typically thousands of potential causal variant models. To make these more interpretable, SNPs in high LD which meet the criteria of substitutability (see Methods) were grouped. The identification of SNP groups is a feature of stochastic search—generally, SNPs in a group have high LD and similar evidence for association, such that a single candidate causal variant is not statistically distinguishable within the group. When we discuss a SNP group model, e.g. model A + B, we mean the collection of models that include exactly one SNP from group A and exactly one SNP from group B, and no others. We consider posterior support for each grouped model (GPP) as the sum of posterior probabilities overall SNP models in that group when interpreting the stochastic search results. SNP group membership is shown in Supplementary Data 3.

While one of the strengths of Bayesian methods is that multiple competing models can be identified with posterior support for each, for the purposes of comparing stochastic search and stepwise search results, we chose to focus on discrepancies between the best models chosen for each. In all regions, the model preferred by stochastic search either had equal or better Bayesian Information Criterion (BIC) and equal or larger number of variants compared to the model chosen by stepwise search (Supplementary Fig. 6). For 16 regions (18 disease-region pairs) the stepwise model was nested in that of stochastic search (treating SNPs in the same SNP group as equivalent; Supplementary Table 2). In six regions (6 disease-region pairs) there appeared to be two separate signals, both weak ($2 \times 10^{-10} < p < 4 \times 10^{-6}$ by single SNP logistic regression) with stochastic search posterior support fairly evenly shared between the two SNP groups, and the SNP selected by stepwise search falls in the group with slightly less posterior support (Table 1). In a further four regions (five disease-region pairs) we found non-nested stochastic/stepwise mismatches, which could not be explained simply.

**Joint tagging of stochastic search models by stepwise SNPs.** We investigated these five mismatch cases further, both mathematically and using simulation, hypothesising that they may reflect cases where the SNP with smallest p-value acts to tag both of two distinct causal variants[17]. We walk through these results using the example of ATD in a chromosome 10p region. Haplotype analysis, which estimates effects for all observed combinations of alleles across these three SNPs, illustrates how the minor allele of stepwise search-selected SNP rs706779 (a member of group J) tends to be carried together with the minor alleles of stochastic search-selected SNPs rs61839660 (group A) and rs11594656 (group C) (Fig. 1a). Considering the haplotypes formed from rs61839660/A, rs706779/J and rs11594656/C, we see that while haplotypes carrying the rs706779:C allele in the presence of either rs61839660:T or rs11594656:A (haplotypes TCT or CCA) are protective for ATD, a haplotype carrying rs706779:C in combination with rs61839660:C and rs11594656:T (CCT, frequency 13%) is indistinguishable from the common (susceptible) haplotype CTT (Fig. 1a, $p = 0.24$, Wald test).

Simulations showed that if the J model (any model with exactly one SNP from group J) were true, both stepwise and stochastic search would correctly identify it (Fig. 1b, Supplementary Table 3). In contrast, if the A + C model (2-SNP model with a SNP from each of groups A and C) were true, stepwise got stuck on J, while stochastic search moved from selecting J at lower sample sizes, to A + C at higher sample sizes (Fig. 1b, Supplementary Table 4, further examples in other regions/ diseases in Supplementary Tables 5–8, SNP group membership in Supplementary Data 3). A small perturbation on the simulated effect sizes for A + C led both methods to select C or A + C directly, indicating that the potential for joint tagging was dependent on the combined effect sizes.

We explored a broader range of combined effect sizes mathematically, finding that there was a high probability of J having the smallest p-value when A and C were causal only when A and C had similar odds ratios; and that our observed data fell

**Table 1 Regions having conflicting models selected by stepwise and stochastic search**

| Region | Disease | SW model | SW P-value | SW model GPP | SW SNP group size | SS model | P-value(s) | SS GPP | SNP group size(s) | LD |
|---|---|---|---|---|---|---|---|---|---|---|
| 2q-100544954-101038647 (AFF3) | iRA | A/rs10209110 | $3.79 \times 10^{-9}$ | 0.427 | 32 | C/rs13415465 | $1.96 \times 10^{-8}$ | 0.514 | 93 | 0.46 |
| 2q-231076289-231235886 (SP110, SP140, SP140L) | iCEL | C/rs6192167 | $8.49 \times 10^{-7}$ | 0.374 | 4 | B/rs12694846 | $3.62 \times 10^{-6}$ | 0.44 | 18 | 0.12 |
| 7p-5024236-50365063 (IKZF1) | T1D | A/rs2168587 | $8.10 \times 10^{-7}$ | 0.123 | 1 | D/rs17552787 | $2.46 \times 10^{-6}$ | 0.457 | 20 | 0.094 |
| 7p-50366637-50694384 (DDC, FIGNL1, GRB10, IKZF1) | T1D | B/rs34046423 | $2.08 \times 10^{-10}$ | 0.226 | 35 | A/rs10264390 | $2.89 \times 10^{-9}$ | 0.657 | 37 | 0.4 |
| 15q-67414055-67469568 (SMAD3) | iCEL | A/rs2289261 | $2.48 \times 10^{-7}$ | 0.324 | 15 | B/rs8024330 | $1.74 \times 10^{-6}$ | 0.342 | 20 | 0.24 |
| 20p-14971197-1689461 (SIRPD/SIRPB1) | T1D | B/rs202535 | $6.79 \times 10^{-9}$ | 0.221 | 9 | C/rs202536 | $1.25 \times 10^{-8}$ | 0.391 | 21 | 0.49 |
| 2q-204446380-204816382 (CTLA4) | T1D | G/rs3087243 | $3.89 \times 10^{-17}$ | 0.00281 | 32 | H/rs231779 + E/rs370078940 | $2.10 \times 10^{-21}$; $1.36 \times 10^{-8}$ | 0.765 | 52; 31 | 0.50; 0.07 |
| 4q-122973062-123565302 (IL2/IL21) | iRA | G/rs3087243 | $1.54 \times 10^{-7}$ | 0.00787 | 32 | H/rs34029700 + E/rs7422494 | $7.04 \times 10^{-9}$; $4.39 \times 10^{-9}$ | 0.753 | 52; 31 | 0.26; 0.05 |
| 4q-122973062-123565302 (IL2/IL21) | T1D | D/rs77516441 | $3.97 \times 10^{-14}$ | 0.0193 | 13 | F/rs13122213 + A/rs6837165 | $1.05 \times 10^{-9}$; $3.43 \times 10^{-16}$ | 0.85 | 53; 106 | 0.20; 0.23 |
| 10p-6030000-6200000 (IL2RA) | ATD | J/rs706779 | $4.63 \times 10^{-8}$ | 0.011 | 2 | C/rs2476491 + A/rs61839660 | $2.89 \times 10^{-9}$; $1.96 \times 10^{-8}$ | 0.954 | 8; 31 | 0.34; 0.13 |
| 14q-101290463-101328739 (MEG3) | T1D | C/rs34552516 | $9.69 \times 10^{-10}$ | 0.0814 | 5 | B/rs1054000 + A/rs11160606 | $1.13 \times 10^{-11}$; $2.36 \times 10^{-6}$ | 0.777 | 5; 16 | 0.29; 0.30 |

Each row summarises results for a single region, defined by chromosome, start and end coordinates (hg19), with neighbouring or previously reported candidate gene names shown for orientation. Each stepwise search (SW) model consists of a single SNP and we also indicate which SNP group it belongs to, by a letter in front of the SNP rs ID; the SNP group size, the p-value of the SNP, and stochastic search group posterior probability (GPP) are also given. Analogous information is given for stochastic search (SS) models and for 2-SNP models the joint p-values from these models are given. The LD column lists the $r^2$ between the stepwise SNP and the SNP(s) from the stochastic search model

within this region (Fig. 1c). A similar pattern was seen at all other mismatch regions (Supplementary Fig. 7).

Finally, we showed that the pattern of LD between three SNPs (two causal and a third tag), together with MAF (minor allele frequency) and effect sizes, determine whether a tag SNP has the smallest expected p-value (Fig. 2a, Supplementary Note 1). At the extremes of this pattern, there is a non-zero probability that the tag model will be erroneously selected, even by a criterion such as BIC which penalises the larger model (Supplementary Note 2). While we cannot identify how many cases of joint tagging may exist in our GWAS data because the causal variants are unknown, we can quantify what proportion of 3 SNP LD matrices match this pattern under an assumption of equal odds ratios at the causal variants. Doing so, we found that 20–40% of potential common causal variant pairs (MAF > 5%) had a potential joint tag, though this was highly variable across regions (Fig. 2b, c, Supplementary Data 4) and should be considered an upper limit because our assumption of equal effect sizes may not be justified.

Together, these results better characterise and quantify the potential frequency of joint tagging, in which a non-causal SNP carried on population haplotypes together with distinct causal SNPs with similar effects may have a smaller single SNP p-value than either causal variant itself. This can cause stepwise search to get stuck on the tag, whereas stochastic search will find both causal variants, if the sample sizes are large enough. With smaller sample sizes, stochastic search may also choose the tag, because such samples may not contain enough information to overcome the strong penalty needed by more complex models to avoid over-fitting. Thus, joint tagging may potentially affect many more cases than identified above by the simple comparison of stepwise and stochastic search results from fixed sample sizes.

**Proposed method for fine-mapping multiple diseases.** We noticed a striking overlap between the fine-mapping results for different diseases in these regions, with 20 of 30 regions with two or more associated diseases showing evidence of overlap (Supplementary Fig. 8), consistent with previous reports of shared genetic etiology between the diseases[2], which inspired the creation of the ImmunoChip. This motivated us to exploit the sharing between diseases, extending the stochastic search approach to jointly analyse multiple diseases, borrowing information between them, to help overcome sample size limitations. We use a multinomial logistic regression framework, the natural extension of the binomial logistic model, where each individual is assumed to belong to exactly one disease group or a pooled group of controls shared between diseases. This formally accounts for the sharing of controls between diseases in different studies.

We introduce the concept of configurations—sets of causal variant models for each disease, and we borrow information between the diseases by means of a prior, which upweights configurations that share one or more causal variants between diseases by a factor $\kappa$ (Fig. 3). Such a parameter is also used in colocalisation analysis, with values ranging from 100[1,18] to 1000[19]. In the case of MFM, it may be easier to elicit a prior on the chance of any sharing in causal variants between a pair of diseases, and we show in Supplementary Note 3 how this value can be used to derive $\kappa$ for two or more diseases. In all our simulations and analyses, we chose $\kappa$ so that the prior on any pair of diseases sharing at least one causal variant in a region where they are both associated is 0.5, compatible with conclusions of previous IMD studies of IMDs[1].

One obvious challenge for dealing with configurations, is that the number of models that needs to be considered for each disease is already large, and the number of possible configurations is the product of these. This implies that exponentially increased

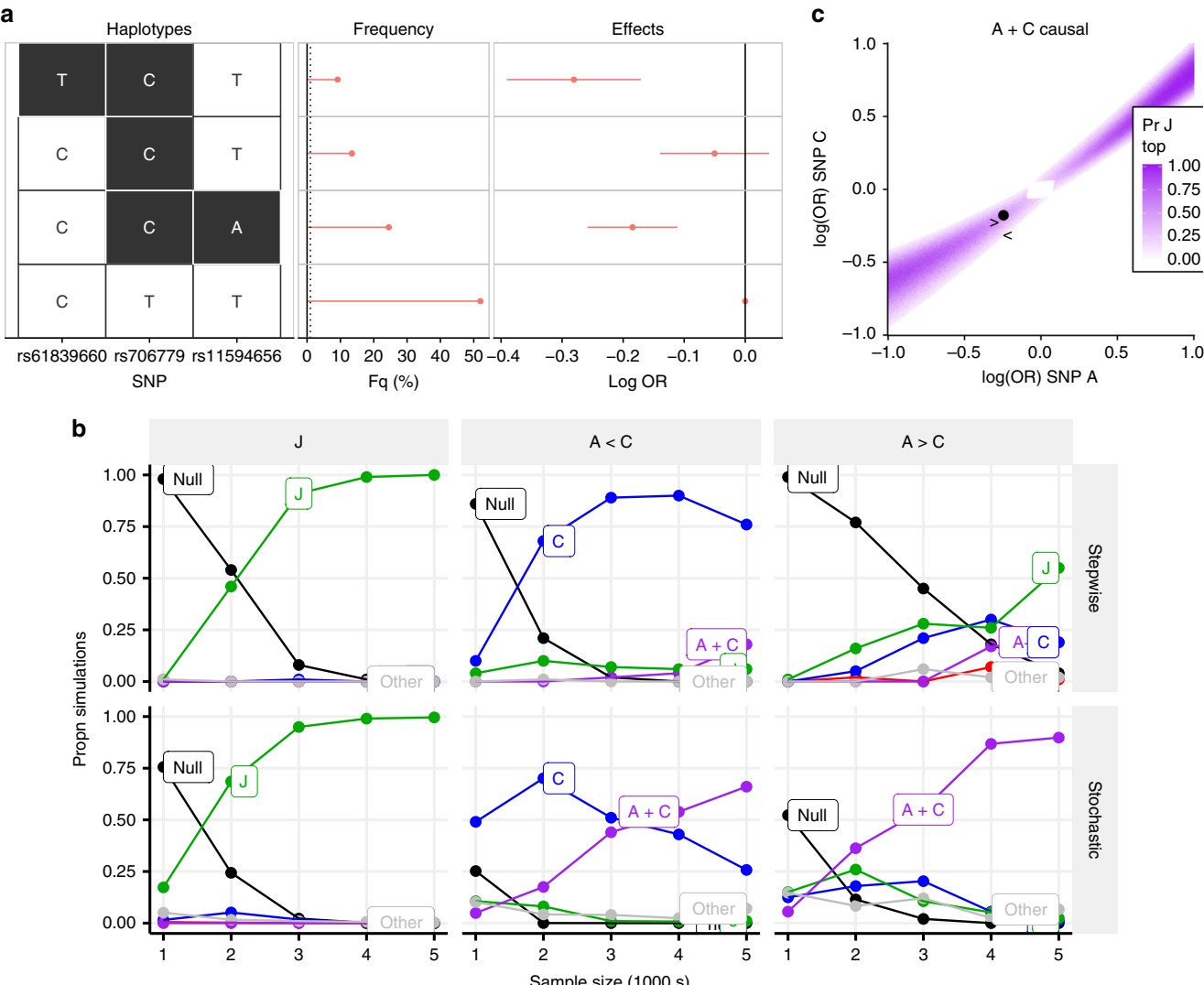

**Fig. 1** Evidence for joint tagging. **a** Haplotype analysis of SNPs selected by stepwise search and GUESSFM for ATD in region 10p-6030000-6220000. A representative SNP from each SNP group is shown. Each column represents one SNP, with possible alleles colour coded according to major or minor. Each row is a haplotype—a specific combination of alleles across all SNPs—with frequency in UK controls and effect on disease risk (log OR + 95% CI). There are four common haplotypes. Three carry the minor allele at the J SNP rs706799, but only those that also carry minor allele at A or C show a significant effect on disease risk. **b** Comparison of stepwise and stochastic search applied to simulated data. Causal variants were simulated as follows: J: single causal variant J, OR = 0.8; A < C causal variants A + C, odds ratios A:0.81, C:0.74; A > C: causal variants A + C, odds ratios A:0.74, C:0.8. Potential models include J (green), C (blue), A + C (purple), A (red) and null (black); any other models are grouped together as grey. The y-axis shows the proportion of simulations in which the stepwise approach chose the indicated model (adding SNPs while $p < 10^{-6}$) or the average posterior probabilities for each model for the stochastic search approach. Sample size (x-axis) is the number of cases and controls. **c** Assuming A and C are causal, this plot shows the probability that J has the smallest p-value as a function of the effect sizes (log odds ratios) at A and C. The estimated effects for A and C from real data are shown by a point, and the simulations from **b** by < and > for A < C and A > C conditions, respectively. Source data for **b** are provided in Supplementary Tables 3–4

computational time and memory will be required to evaluate all configurations, and to store these results. We provide solutions for both challenges. First, we show the log Bayes factor for a multinomial model that simultaneously considers all diseases can be approximated by a quantity that can be rapidly calculated—the sum of the log Bayes factors for the corresponding logistic models for each individual disease and an offset term determined by sample and model sizes (Supplementary Note 3). Second, we show that the marginal (single disease) model posteriors from the multinomial model can be calculated without needing to store the individual configuration Bayes factors (Supplementary Note 3). These insights solve both computational time and memory challenges: joint analysis of 2–6 diseases in the *IL2RA* region, (after individual stochastic search results were generated with

GUESSFM), takes only 15–83 s. We can deal with multiple populations, with not all populations represented for all diseases, by noting that when controls are not shared, the joint log Bayes factor is a simple sum of logistic log Bayes factors, allowing us to fit a multinomial to the samples from common populations with shared controls, and add disease-specific log Bayes factor terms from logistic models fitted to the distinct populations.

Finally, to enable interpretation of the posterior probability of thousands of models for each disease, which typically contain many models differing only by the exchange of one SNP for another in high LD, we formalise the method for grouping SNPs across multiple diseases by hierarchical clustering of SNPs according to their LD ($r^2$) and the probability of being jointly required to explain disease, grouping SNPs selected with some

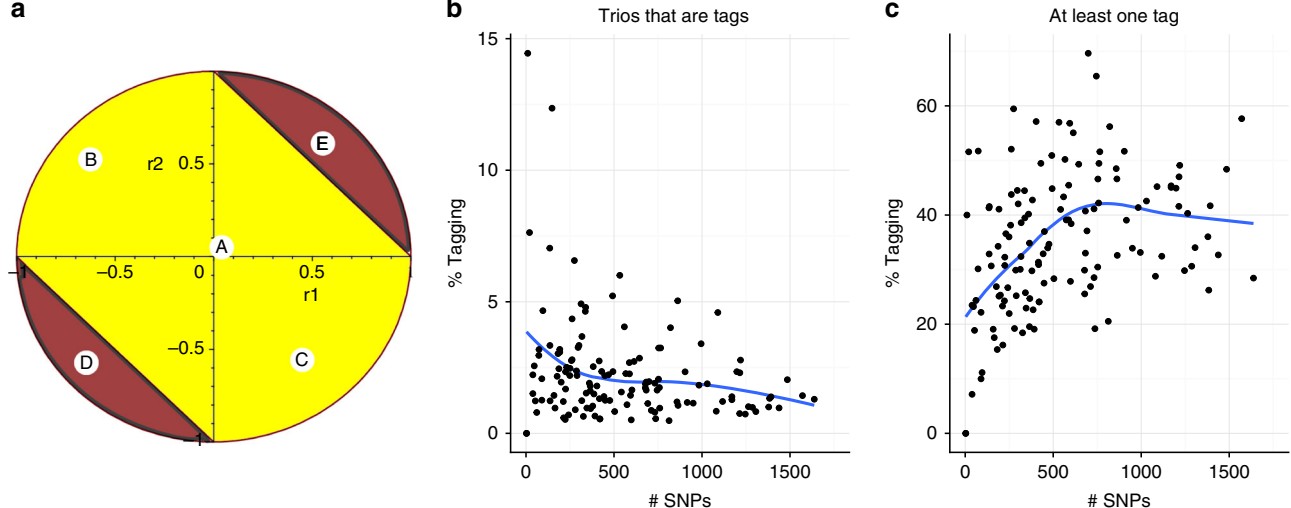

**Fig. 2** Potential frequency of joint tagging. We consider the patterns of three-way LD between each possible trio of SNPs, nominating the first two as causal, and the third as a potential tag. **a** For each pair of potential causal SNPs, we can predict whether the third SNP is a tag according to the pairwise correlation between that SNP and the two potentially causal SNPs (r1, r2). Red (yellow) areas indicate settings where the third SNP is (is not) a potential tag for SNPs 1 and 2. In this example, the potentially causal SNPs have equal MAF, equal effect on disease risk (equal odds ratios, OR) and are uncorrelated. Then, if the third SNP is (A) uncorrelated or weakly correlated with either SNP 1 or 2, or (B, C) negatively correlated with one and positively with the other, we would not expect it to act as a tag. On the other hand, if it were (D, E) strongly positively or negatively correlated with both causal variants, we would expect it to act as a joint tag. **b** shows the result of searching all possible SNP trios in UK ImmunoChip control data, and quantifying the proportion of trios that correspond to joint tagging in each region, assuming the causal variants have equal OR; the pattern is individually rare, consistently <5%. **c** shows the proportion of SNP pairs for which at least one potential tag exists, which can be substantial—about 40% overall. The blue curves show the loess fits to the points. Source data are provided in Supplementary Data 4

nominal posterior probability, which are in high LD and rarely selected together in any model (Methods).

**MFM increases chance of selecting the correct model**. We examined the performance of MFM by simulation. We found that when causal variants overlapped between diseases, MFM was able to recover the correct models at smaller sample sizes than individual disease analysis (Fig. 4a, b, Supplementary Data 5, 6), i.e. sharing information between diseases contributed to a gain in accuracy similar to increasing sample size for each disease. When no causal variants were shared, multinomial and independent approaches gave similar results (Fig. 4c, Supplementary Data 7), i.e. sharing information did not tend to mislead as long as there were strong signals in each disease. When one disease had no causal variants, multinomial and independent results were again similar (Fig. 4d, Supplementary Data 8); i.e. no information is gained but there is also no noticeable loss in accuracy in doing so.

**MFM analysis of up to six IMD**. We applied MFM to all 30 ImmunoChip regions with at least two associated diseases (Supplementary Data 9, visualised at https://chr1swallace.github.io/ MFM-output/index.html). We identified seven regions for which the top model by independent stochastic search and MFM differed (Table 2). Four of these were single SNP models under independent analysis, which moved to an alternative single SNP in MFM. For three of these four, the difference was seen in analysis of a UK-only subset, so that we could consider independent analysis of the UK + international data, which included more samples but used the more conventional analysis method as an adjudicator. In all three cases, this adjudicator matched the MFM analysis of the UK-only data, suggesting that UK independent analysis was limited by power, and that UK MFM analysis increased power, allowing conclusions to be drawn that were consistent with those seen in a larger single disease analysis.

One of the multi-SNP regions that showed differences across multiple diseases was on chromosome 2q, harbouring the candidate gene *CTLA4*. In stepwise analysis, iRA, T1D, ATD and CEL all converge on a single SNP model, in the group labelled G in the stochastic search results, while for iCEL a single SNP is selected in group I (Table 3, Fig. 5a). For single disease stochastic search, we find CEL (UK-only) and ATD have a single signal in the group labelled G, matching the stepwise results, while RA and T1D both have two signals, in groups labelled E and H, represented by causal variant configuration E + H. The iCEL result is more uncertain, with the posterior spread between I + K, I or E + G. Note that K is also the second selected SNP for iCEL stepwise regression ($p = 4 \times 10^{-6}$), although it doesn't reach our adopted significance threshold. Simulations show that G may tag an E + H model (Fig. 5b-c, Supplementary Tables 5–6, SNP group membership in Supplementary Data 3).

MFM finds increased support for E + H for RA and T1D while the CEL and iCEL results become more concentrated with support for G or E + G (Table 3). While we suggested G may tag E + H, MFM maintains strongest support for G in ATD, although there is also posterior support for H in combination with other groups (group marginal posterior probability of inclusion, gMPPI = 0.60). A previous attempt to fine-map autoimmune disease association, by colocalisation analysis of T1D, RA and CEL (using the same UK data as here) came to similar conclusions, finding strong support for E + H models for iRA and T1D and either G or E + G for CEL[1]. However, a more recent analysis of T1D and RA, also in largely the same samples, identified a different pair of variants, rs3087243 (G) and rs117701653 (C)[20] for both diseases using an exhaustive search of all one and two SNP models.

We compared the models suggested by all these studies across all diseases by BIC (Supplementary Data 10) and using haplotype analysis (Fig. 5d). This visually highlighted rs117701653/C identified for iRA by exhaustive search[20] and rs76676160/K

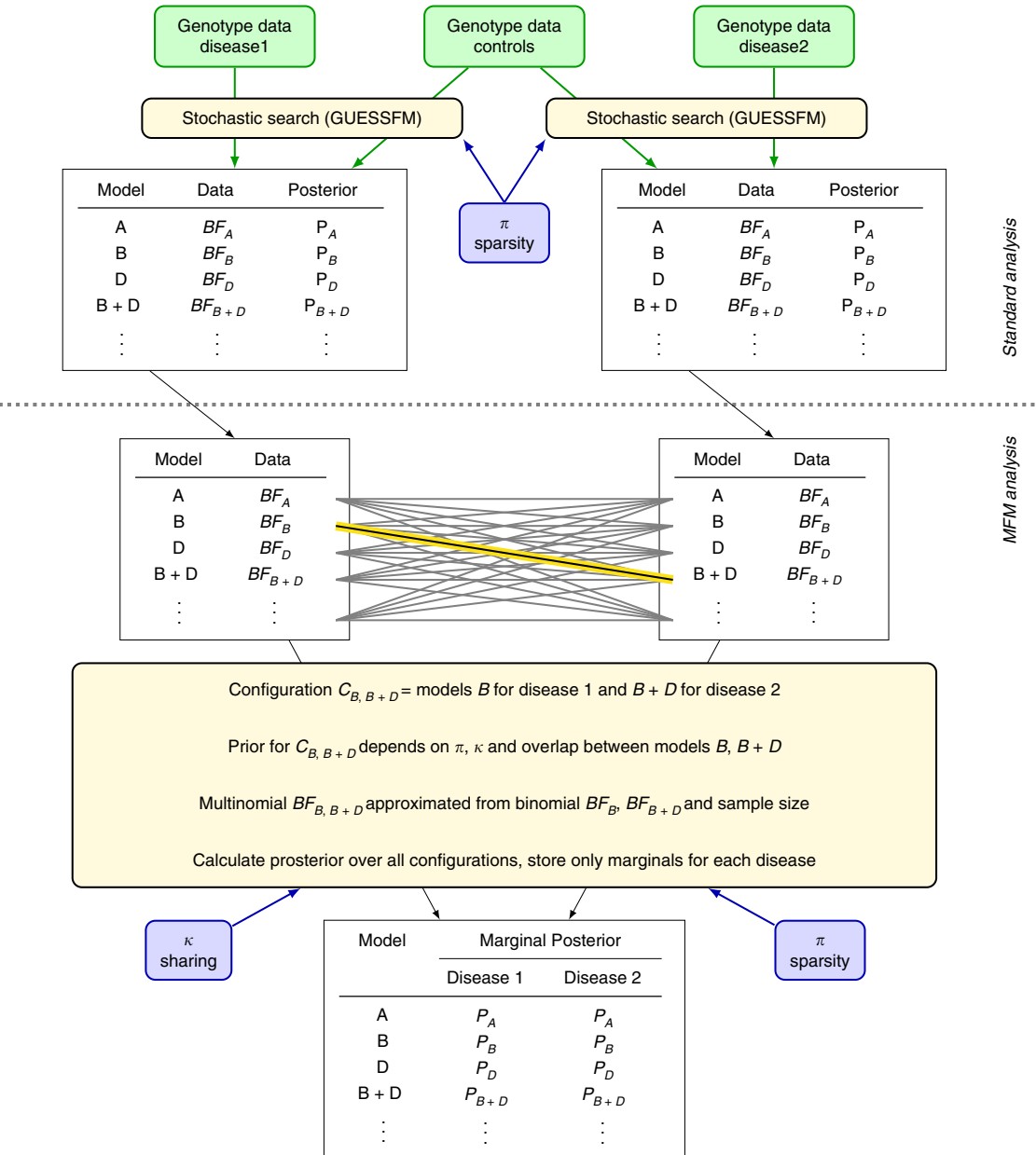

**Fig. 3** Schematic diagram for Multinomial Fine-Mapping (MFM) method. MFM is used for multiple diseases with shared controls and, for simplicity, only two diseases are shown. After selection of a sparsity prior parameterised by π, stochastic search is applied individually to each disease, as in standard analyses. Marginal results are combined in MFM, using an approximation to the multinomial Bayes factor (BF) and with information shared between diseases using a sharing prior, which upweights joint models with shared causal variants by a factor κ. Memory constraints are dealt with by storing only the marginal posterior distributions for each disease

identified by stochastic search for CEL and iCEL as having similar protective effects across all diseases and low minor allele frequencies (<0.05). The two SNPs are unlinked ($r^2 < 0.01$) and in low LD with other genotyped or imputed SNPs outside their groups ($r^2 < 0.2$). The 2-SNP models E + H identified here, and G + C[20] have similar BIC in our data for iRA and iCEL (Supplementary Data 10), but the greater number of SNPs in the E and H groups mean that E + H encompasses many more possible causal variant pairs and so has greater grouped posterior support. Additionally, individual E + H models have a clearly better fit than G + C for T1D (Supplementary Data 10). In total, results in this region exemplify the difficulty with fine-mapping multiple causal variants in the presence of complex LD, and suggest the region likely contains three common causal variants,

in groups E (CEL, RA, T1D), G (CEL and ATD) and H (RA and T1D, and possibly ATD) and possibly two low frequency causal variants in groups C and K (RA, CEL).

Our previous report of stochastic-stepwise mismatch focused on MS and T1D in the *IL2RA* region[9]. We identified four groups of SNPs corresponding to four causal variants for T1D, with results agreeing between stepwise and stochastic search[9]. However, while stepwise search identified a single SNP for MS, rs2104286 (group B), stochastic search identified two distinct variants in groups A and D (posterior probability 55%), and suggested that rs2104286/B was a joint tag for these groups ($r^2 = 0.334$ and 0.301, respectively)[9], a conclusion supported by haplotype analysis and simulations here (Fig. 6, Supplementary Tables 7, 8).

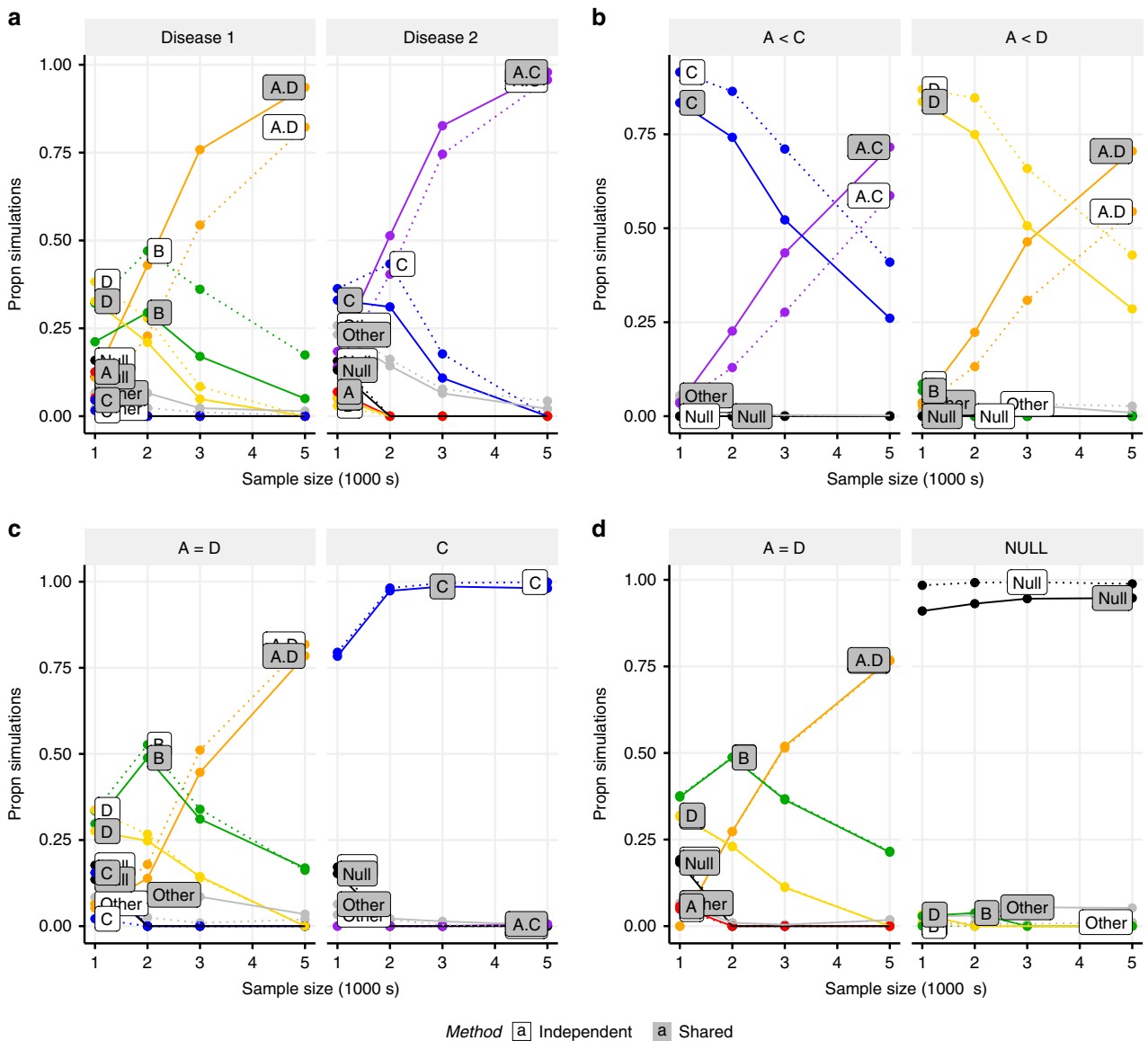

**Fig. 4** Comparison of MFM analysis and single disease analysis. Causal variants were simulated for two diseases with models defined by SNP groups from the *IL2RA* region. MFM is shown by solid lines and independent analyses by dashed lines. Throughout, disease 1 has causal variants A + D, while causal variants for disease 2 vary. **a, b** Disease 2 has causal variants A + C and the odds ratio of A, $OR_A$, is the same for both diseases; **a** A has a stronger effect than C and D; $OR_A = 1.4$ (both), $OR_D = 1.25$ (disease 1), $OR_C = 1.25$ (disease 2). **b** A has a weaker effect than C and D; $OR_A = 1.25$ (both), $OR_D = 1.4$ (disease 1), $OR_C = 1.4$ (disease 2). **c** Disease 2 has only C causal; $OR_A = OR_D = 1.25$ (disease 1), $OR_C = 1.25$ (disease 2). **d** Disease 2 has no causal variants (no association). Potential models include A (red), B (green), C (blue), D (yellow), A + D (orange), A + C (purple) and null (black); any other models are grouped together as grey. The y-axis shows the average posterior probabilities for each model. **a, b** MFM can identify the true two causal variant model at smaller sample sizes than independent analysis in simulated data when there is sharing between diseases. **c, d** When there is no sharing (**c**) or one disease has no true associations (**d**), no information is gained by using MFM but there is only minimal loss in accuracy in doing so. Source data are provided in Supplementary Data 5–8

While our previous analysis included UK and non-UK (international) cases and controls for MS, here we used only the UK subset, and both stepwise and stochastic search identified B (group posterior probability, GPP = 0.632), with the A + D model having only GPP = 0.188, consistent with results that stochastic selection of a joint tag depends on sample size (Table 3). A more recent stepwise analysis of a larger, international sample has identified two SNPs, rs11256593 and rs12722559[21]. rs12722559 ($r^2$ 0.323 with rs2104286/B) is in our group H (GPP = 0.114, third strongest stochastic search model) while rs12722559 ($r^2$ 0.482 with rs2104286/B, MPPI = $1.20 \times 10^{-5}$) was not in our SNP groups. In

our UK data, we found the best fitting models were A + D (BIC 19299.46) and B (19302.88), both significantly better fits than rs11256593 + rs12722559 (BIC 19320.06).

For ATD, stepwise search identified a 1-SNP model, rs706779/ J, consistent with previous analyses of ATD[13,22], and matching the top reported SNP for another IMD, Vitiligo[23], while stochastic search selected a two SNP model, A + C (Table 1, Supplementary Data 11). MFM maintained support for the A + C model for ATD, and preferred the 2-SNP A + D model for MS (Table 3), agreeing with our previous stochastic search results for a larger UK + international MS dataset[9]. Limited power may also

**Table 2 Regions with conflicting models chosen by independent disease analysis and MFM**

| Region | Disease | Other diseases | Independent | MFM | Mean $r^2$ between groups |
|---|---|---|---|---|---|
| 1p-2406887-2785671 (MMEL1, TNFRSF14) | RA | CEL, MS | D/rs4648662 | C/rs10752749 | 0.36 |
| 1p-2406887-2785671 (MMEL1, TNFRSF14) | iRA | iCEL, MS | C/rs141426426 | C/rs10797431 | 1 |
| 6q-90806835-91039808 (BACH2) | RA | ATD, T1D | G/rs56258221 | C/rs72928038 | 0.33 |
| 6q-90806835-91039808 (BACH2) | iRA | ATD, T1D | C/rs72928038 | C/rs72928038 | 1 |
| 18p-12738413-12924117 (PTPN2) | CEL | T1D | F/rs34799913 | C/rs12967678 | 0.4 |
| 18p-12738413-12924117 (PTPN2) | iCEL | iRA, T1D | C/rs67878610 | C/rs12967678 | 1 |
| 7p-37363978-37440453 (ELMO1) | MS | CEL | A/ rs1962401 | C/rs77801025 | 0.47 |
| 2q-204446380-204816382 (CTLA4) | iCEL | ATD, iRA, T1D | I/rs2162610 + K/rs76676160 | G/rs3087243 + E/rs3116499 | (I,G): 0.14 (I,E): 0.17 (K,G): 0.031 (K,E): 0.004 |
| 10p-6030000-6220000 (IL2RA) | MS | ATD, iRA, T1D | B/rs2104286/ | A/rs12722496 + D/rs7089861 | 0.2 0.3 |
| 16p-11017058-11307024 (DEXI) | MS | T1D | A/rs11643622 | B/rs12708716 + D/rs4780346 | 0.3 0.3 |

Each row summarises results for a single region, defined by chromosome, start and end coordinates (hg19), with a previously reported candidate gene name shown for orientation. The best model for each method is selected by group posterior probability (GPP) and for each method the best SNP models for each group(s) are given as representatives of the group models. The last column gives the mean $r^2$ between the SNP group(s) of independent analyses and those of MFM. The other diseases that were used in MFM are listed under Other Diseases

**Table 3 Summary results for fine-mapping in *CTLA4* and *IL2RA***

| Region | Disease | SW model | SW P | Indep. model | Indep. PP | MFM (UK) model | MFM (UK) PP | MFM (Int.) model | MFM (Int.) PP |
|---|---|---|---|---|---|---|---|---|---|
| *CTLA4* | ATD | G/rs11571297 | $1.22 \times 10^{-24}$ | G | 0.842 | G<br>G + H | 0.593<br>0.349 | G<br>H + I<br>G + H<br>E + H | 0.374<br>0.273<br>0.236<br>0.102 |
| | CEL | G/rs3087243 | $1.48 \times 10^{-12}$ | G<br>G + K | 0.641<br>0.136 | G<br>E + G | 0.517<br>0.281 | | |
| | iCEL | I/rs2162610 | $3.74 \times 10^{-14}$ | I + K<br>I<br>E + G | 0.351<br>0.14<br>0.115 | | | E + G | 0.829 |
| | iRA | G/rs3087243 | $1.54 \times 10^{-7}$ | E + H<br>A + E + H | 0.753<br>0.142 | | | E + H | 0.805 |
| | T1D | G/rs3087243 | $3.89 \times 10^{-17}$ | E + H | 0.765 | E + H<br>G | 0.687<br>0.135 | E + H | 0.904 |
| *IL2RA* | ATD | J/rs706779 | $4.63 \times 10^{-8}$ | A + C | 0.954 | A + C | 0.985 | A + C | 0.986 |
| | MS | B/rs2104286 | $1.13 \times 10^{-13}$ | B<br>A + D<br>H | 0.632<br>0.188<br>0.114 | A + D | 0.883 | A + D | 0.901 |
| | iRA | I/rs706778 | $7.55 \times 10^{-8}$ | I | 0.966 | | | I<br>A | 0.695<br>0.201 |
| | T1D | A/rs61839660<br>C/rs11594656<br>E/rs12220852 | $3.60 \times 10^{-34}$<br>$5.85 \times 10^{-12}$<br>$8.79 \times 10^{-10}$ | A + C + E + F<br>A + E + F + H + I | 0.622<br>0.201 | A + C + E + F<br>A + E + F + H + I | 0.684<br>0.178 | A + C + E + F<br>A + E + F + H + I | 0.674<br>0.191 |

For each disease, the following are provided: selected stepwise (SW) model and conditional SNP p-values, high PP models (and PP) for each of independent analyses (Indep.), MFM (UK samples only) and MFM with international samples. *CTLA4* and *IL2RA* are the regions 2q-204446380-2048163 and 10p-6030000-6220000, respectively

affect RA-international in this region, for which individual analysis picked group I (97%) and MFM support was split between groups A (20%) and I (70%).

Our results emphasise the importance of the A group, which is selected for three of the four diseases (T1D, MS and ATD). This group of SNPs have been previously associated with variation in the expression of *IL2RA* mRNA and of its encoded protein, CD25, in CD4$^+$ memory T cells[24,25], and a recent allele-specific expression (ASE) study has pinpointed the causal variant affecting mRNA expression among the set as rs61839660[26]—notably the same variant identified in an IBD GWAS of 67,852 individuals[27] and an eczema/dermatitis phenotype in a GWAS of thousands of phenotypes for 337,000 samples in the UK Biobank[28], supporting the notion that this SNP has a common effect across multiple diseases that is missed in stepwise analysis of MS and ATD. We note that the direction of effect for rs61839660 is opposite in IBD and eczema/dermatitis (risk allele T) compared to T1D, MS, ATD and JIA[15] (risk allele C). We note also that the minor alleles of group I SNPs (represented by

rs706778 and rs11256557 in the haplotype analysis, Supplementary Fig. 9) selected for RA-international are carried along with the minor protective alleles of groups A, C and D and it is possible that the group I SNPs are tagging three *IL2RA* SNP groups.

**Allele-specific expression confirms functional effects.** In addition to linking group A SNPs to *IL2RA* expression, we have shown that SNPs in group D decrease the percentage of CD25 expressing naive T cells[9,24]. Here, we extend our analysis of *IL2RA* mRNA expression to examine any effects of rs2104286/B in the context of groups A and D. ASE assays compare relative expression between paternally and maternally inherited chromosomes in individuals heterozygous for a putative functional SNP according to the allele each chromosome carries at the SNP. It is a powerful design, because the within-individual comparison controls for between individual biological variation resulting from other genetic and environmental differences. We quantified ASE of *IL2RA* mRNA in memory and naive CD4$^+$ T cells

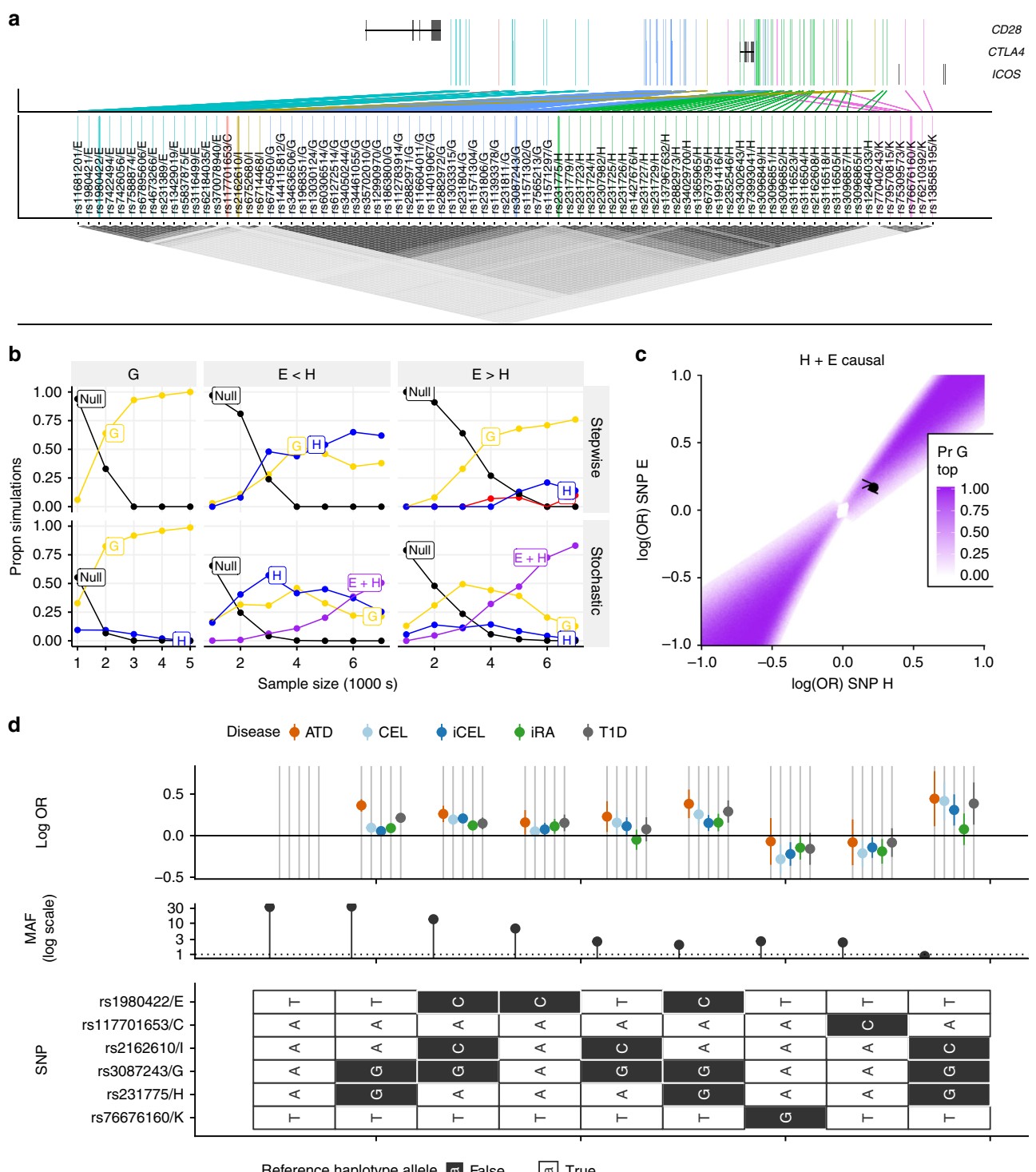

**Fig. 5** Analysis of chromosome 2q region containing *CTLA4*. **a** Map showing positions of SNPs (GRCh37) colour coded by SNP group. SNPs in the same group are in high LD. **b** Comparison of stepwise and stochastic search applied to simulated data. Causal variants were simulated as follows: G: causal variant G, OR = 1.25; E < H and E > H causal variants E + H, E < H:OR$_E$ = 1.19, OR$_H$ = 1.24 (observed in T1D data); E > H: OR$_E$ = 1.24, OR$_H$ = 1.19. Possible models include E (red), G (yellow), H (blue), E + H (purple) and null (black); any other models are grouped together as grey. The *y*-axis shows the proportion of simulations in which the stepwise approach chose the indicated model (adding SNPs while *p* < 10$^{-6}$) or the average posterior probabilities for each model for the stochastic search approach. Sample size (*x*-axis) is the number of cases and controls. **c** Assuming E and H are causal, this plot shows the probability that G has the smallest p-value as a function of the effect sizes (log odds ratios) at E and H. The estimated effects for E and H from T1D data are shown by a point, and the simulations from **b** by < and > for E < H and E > H conditions, respectively. **d** Haplotype analysis of SNP groups with support in any analysis. Each row represents one SNP, with possible alleles colour coded according to major or minor. Each column is a haplotype—a specific combination of alleles across all SNPs—with frequency in UK controls and effect on disease risk (log OR + 95% CI). MAF is shown as a percentage on a log scale to allow frequencies of rarer haplotypes to be distinguished. Source data for **b** are provided in Supplementary Tables 5, 6

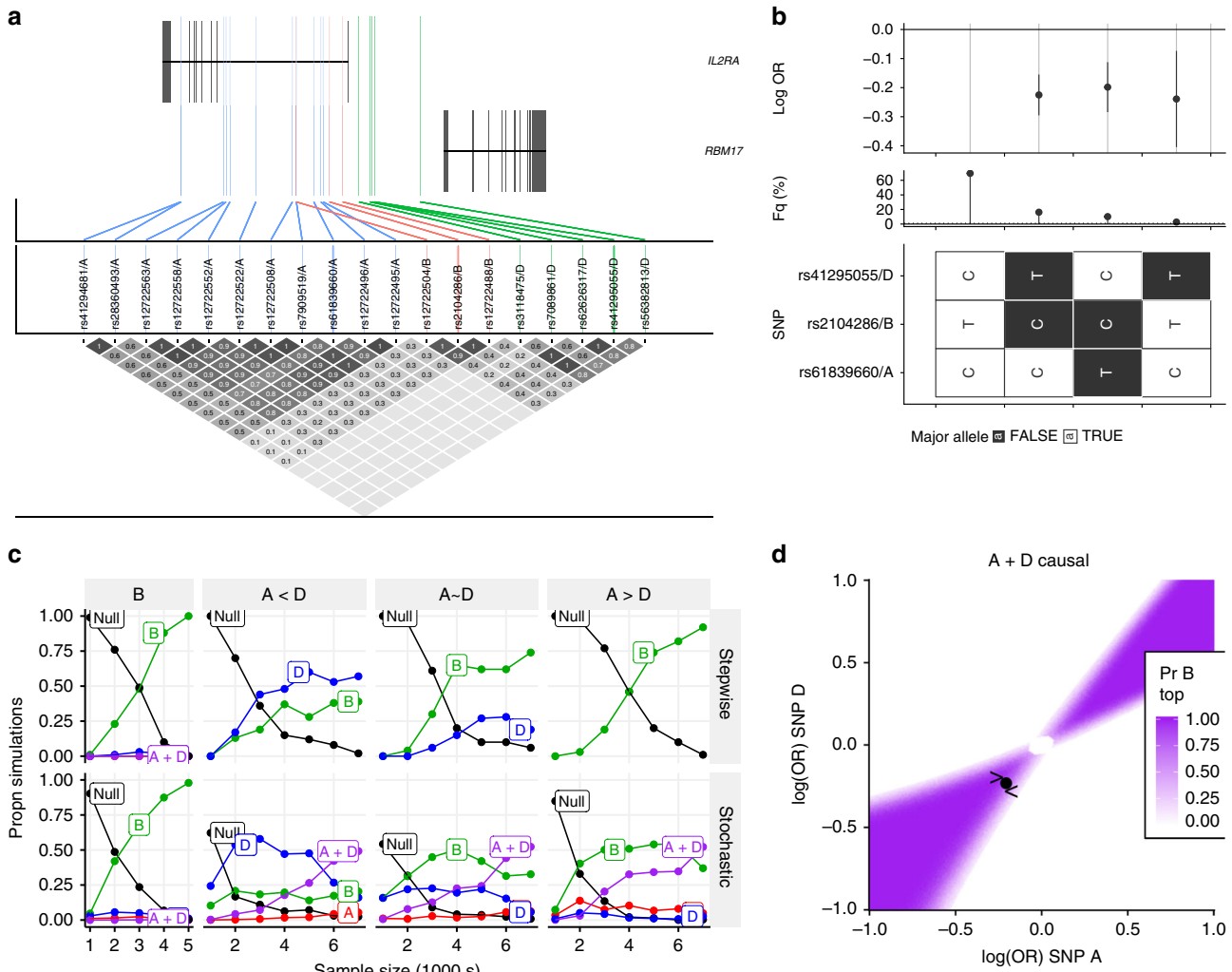

**Fig. 6** Analysis of chromosome 10p region containing *IL2RA* **a** Map showing positions of SNPs (GRCh37) in groups A, B and D. SNPs in the same group are in high LD, with colour used to indicate group membership. **b** Haplotype analysis of SNPs selected by stepwise search and GUESSFM for MS. Each row represents one SNP, with possible alleles colour coded according to major or minor. Each column is a haplotype—a specific combination of alleles across all SNPs—with frequency in UK controls and effect on disease risk (log OR + 95% CI). There are four common haplotypes. Three appear protective, carrying the minor allele at either A or D, but only two carry the minor allele at B. **c** Comparison of stepwise and stochastic search applied to simulated data. Causal variants were simulated as follows: B: single causal variant B, OR = 0.8; A < D causal variants A + D, odds ratios A:0.84, D:0.77; A~D: causal variants A + D, odds ratios A:0.81, D:0.8 (observed in MS data); A > D: causal variants A + D, odds ratios A:0.77, D:0.84. Potential models include A (red), B (green), D (blue), A + D (purple) and null (black); any other models are grouped together as grey. The *y*-axis shows the proportion of simulations in which the stepwise approach chose the indicated model (adding SNPs while $p < 10^{-6}$) or the average posterior probabilities for each model for the stochastic search approach. Sample size (*x*-axis) is the number of cases and controls. **d** Assuming A and D are causal, this plot shows the probability that B has the smallest p-value as a function of the effect sizes (log odds ratios) at A and D. The estimated effects for A and D from MS data are shown by a point, and the simulations from **c** by < and > for A < D and A > D conditions respectively. Source data for **c** are provided in Supplementary Tables 7, 8

isolated from 36 donors selected by genotype from a bioresource (www.cambridgebioresource.org.uk) to be heterozygous at SNPs in group A (A-het), D (D-het) or both (A + D-het). To control for other potential effects, we chose donors homozygous for SNPs in groups C and F. The pattern of LD in the region means that the large majority of A-het and D-het individuals are also heterozygous at the B SNP and A + D-het individuals are homozygous at the B SNP (Fig. 7a, Supplementary Data 12), allowing us to directly compare the effects of SNPs in groups A, B and D.

In memory CD4+ T cells, A-het and A + D-het individuals showed an allelic imbalance with the MS-protective A haplotype producing more *IL2RA* mRNA, inconsistent with B causing the imbalanced expression since A + D-het individuals tested are homozygous for B (Fig. 7b). Also inconsistent with B causality is

the lack of allelic imbalance in memory T cells from D-het individuals who are heterozygous at B. In naive CD4+ T cells, D-het as well as A + D-het heterozygotes had an allelic imbalance with the protective D haplotype producing less *IL2RA* mRNA than the susceptible or protective A haplotypes, confirming our previous observations of decreased CD25+ naive CD4+ T cells associated with donors having the protective D haplotype[9]. Again, this is inconsistent with B causality, since only D-het and not A + D-het individuals are heterozygous at B. In A-het donors there appears to be an allelic imbalance in naive CD4+ T cells favouring the MS-protective versus susceptible haplotype, which is the opposite direction to that observed with protection at D and could reflect an anticipatory differentiation of naive T cells toward the memory lineage and its phenotype of increased CD25

expression in A haplotype donors. However, it is not significant, and we did not observe an increase in CD25[+] naive T cells associated with the MS-protective A allele in a previous study[24].

Additionally, we identified four individuals, three of whom carry rare *IL2RA* haplotypes (Fig. 7c): donor 1 carries a common haplotype combination that is homozygous across A, B, D; donor 2 carries the minor allele at B in the absence of a minor allele at either A or D, donor 3 carries a minor allele at D but not B, and donor 4 also carries a minor allele at D but not at B on one haplotype and minor alleles at A and B on the other haplotype (Fig. 7a). Neither donor 1 or 2 demonstrated an ASE in either the memory or naive T cells, an expected result for donor 1 who does not carry a minor allele at A, D or B, and a result from donor 2 showing that the minor allele at B is not associated with either phenotype. ASE results from donors 3 and 4 were consistent with those of D-hets and A + D hets, respectively, shown in Fig. 7b, even though the status of the B SNP was different. These rare donors are consistent with our conclusions that differences seen in *IL2RA* mRNA expression are controlled by the A and D SNPs, in memory and naive CD4[+] T cells, respectively, and argue that the B SNP tags two functionally distinct groups of SNPs, A and D.

## Discussion
Fine-mapping is a general problem in statistical genetics, important in its own right and for informing integrative downstream analyses[19,29]. We have shown that there are candidate causal SNP models for which stepwise regression does not converge to the correct solution, even with increasing sample size, and described the constraints on LD that give rise to this joint tagging phenomenon. In principle, exhaustive search could overcome the problem, but scalability is a substantial problem: in a 1000-SNP region: there are a manageable 0.5 million 2-way models but over 166 million 3-way models (and 41.5 billion 4-way models), which cannot be fit in reasonable time for a logistic model that requires optimisation for each model. This exponential growth of model numbers has motivated different approaches to scalable search strategies using linear models and exhaustive search[6,30] or specialised search strategies[7,8,31]. We show that a logistic model stochastic search[9] is feasible and does tend to the correct solution as sample sizes increase. However, even stochastic search methods are limited by existing sample sizes when there are multiple causal variants in proximity, and may produce similar results to stepwise methods when sample sizes are insufficient. MFM could be easily adapted as a wrapper around any of the linear model methods above, provided that the linear model is considered an acceptable approximation to a logistic model and that controls are either shared completely or not at all.

MFM borrows information across diseases and is thus related to, but distinct from, methods that compare two[19] or more[32] traits, which integrate over the fine-mapping posteriors of individual traits, upweighting models that share causal variants, to determine whether there is evidence for sharing. Here, we exploit a prior belief that traits studied are enriched for colocalisation to determine the marginal fine-mapping posterior for each trait, and remove the common colocalisation assumptions of independent datasets and a single causal variant per trait in any region. We also avoid enforcing identity of causal SNPs or their effect sizes between different diseases, as in analysis of an overarching disease phenotype (e.g. autoimmune disease[20]). It is clear from our results that, causal variants may differ between diseases in the same region and that, even when causal variants are shared, magnitude and direction of effects may differ between diseases. MFM could be applied to other collections of diseases where causal variants may tend to be shared, such as psychiatric

diseases[33] or metabolic-related traits[34] if appropriate priors can be elicited for each collection. This might be possible from prior work as here, but if not we recommend that a range of plausible values be considered, with robust results identified as those which remain similar under different priors.

One key result from our analysis is that sample sizes in the low tens of thousands may still not be large enough to robustly fine-map multiple causal variants. This motivates continued collection of GWAS samples for diseases too infrequent to be found in large numbers in the Biobank style datasets, and greater sharing of data between researchers working on related diseases to better map the most likely genetic causal variants. A particular note of caution is raised by the genomic locations where we find discrepancies between stochastic and stepwise results. These are almost entirely those with the strongest biological prior for involvement in these diseases, and also those with typically the strongest effects, and thus greatest power. We question whether these regions are most likely to give rise to discrepancies because they harbour the largest numbers of potential effects or whether, if we had access to much larger datasets, we would see similar discrepancies genome-wide.

Our analysis of six diseases reveals several cases where there appear to be multiple functional haplotypes—i.e. more than one IMD causal variant in a region—that affect different diseases differently. Thus, these functional haplotypic maps are essential for designing biological follow-up experiments, for which we need to decide not just what variants to test, but also what variants to hold constant to avoid confounding the effect of the variant of immediate interest. Note that in our ASE work, testing of B heterozygotes, which are in fact a 2:1 mix of D heterozygotes and A heterozygotes, would have resulted in bimodal results in both the memory and naive CD4[+] T cells subsets. The ability of stochastic search to suggest alternative models provides us the knowledge to compare such models biologically, thereby allowing homogeneous phenotypic groups to emerge that were differently associated with the A and D SNP groups. Our approach can be expanded in a haplotype-directed manner to other accessible immune cell types to determine cell-specific and activation-specific influences of each disease-associated SNP group (A, C, D, E and F) on *IL2RA* mRNA expression, enabling a more complete picture of how particular haplotypes mediate protection or susceptibility to disease. The association of the minor alleles of the A haplotype with disease protection for T1D, MS and ATD, but with disease susceptibility for eczema and IBD, could be caused by A-mediated regulation of *IL2RA* expression in two different cell types: one critical for T1D, MS and ATD disease pathogenesis, the other type pivotal for eczema and IBD. Alternatively, the genetically-determined level of CD25 on memory CD4 T cells could influence their likelihood of differentiating into particular types of cytokine-producing effector cells, a phenotype beneficial for some diseases but not others. We propose that, rather than attempting to colocalise eQTL signals and disease associations that are both determined by stepwise analysis[35], disease haplotype-directed searches for allele-specific expression exemplified in this study will lead to greater clarity when unraveling cellular mechanisms in immune-based diseases.

## Methods
**Simulations—single trait**. Simulations were carried out under a realistic scenario that mimics the MAF and $r^2$ in the *IL2RA* region. We simulated haplotypes for 345 SNPs in chromosome 10p-6030000-6220000 (GRCh37/hg19), based on the CEU 1000 Genomes Phase 3 data[36] (ftp://ftp.1000genomes.ebi.ac.uk/vol1/ftp/release/20130502/) using HapGen2[37]. Code to perform the simulations can be found in https://github.com/jennasimit/MFMextra. Causal variants were selected within SNP groups for each disease model (see Supplementary Data 11) with various OR relating the odds of disease in heterozygote carriers of the non-reference allele compared to the homozygote reference allele. We assumed a multiplicative model throughout.

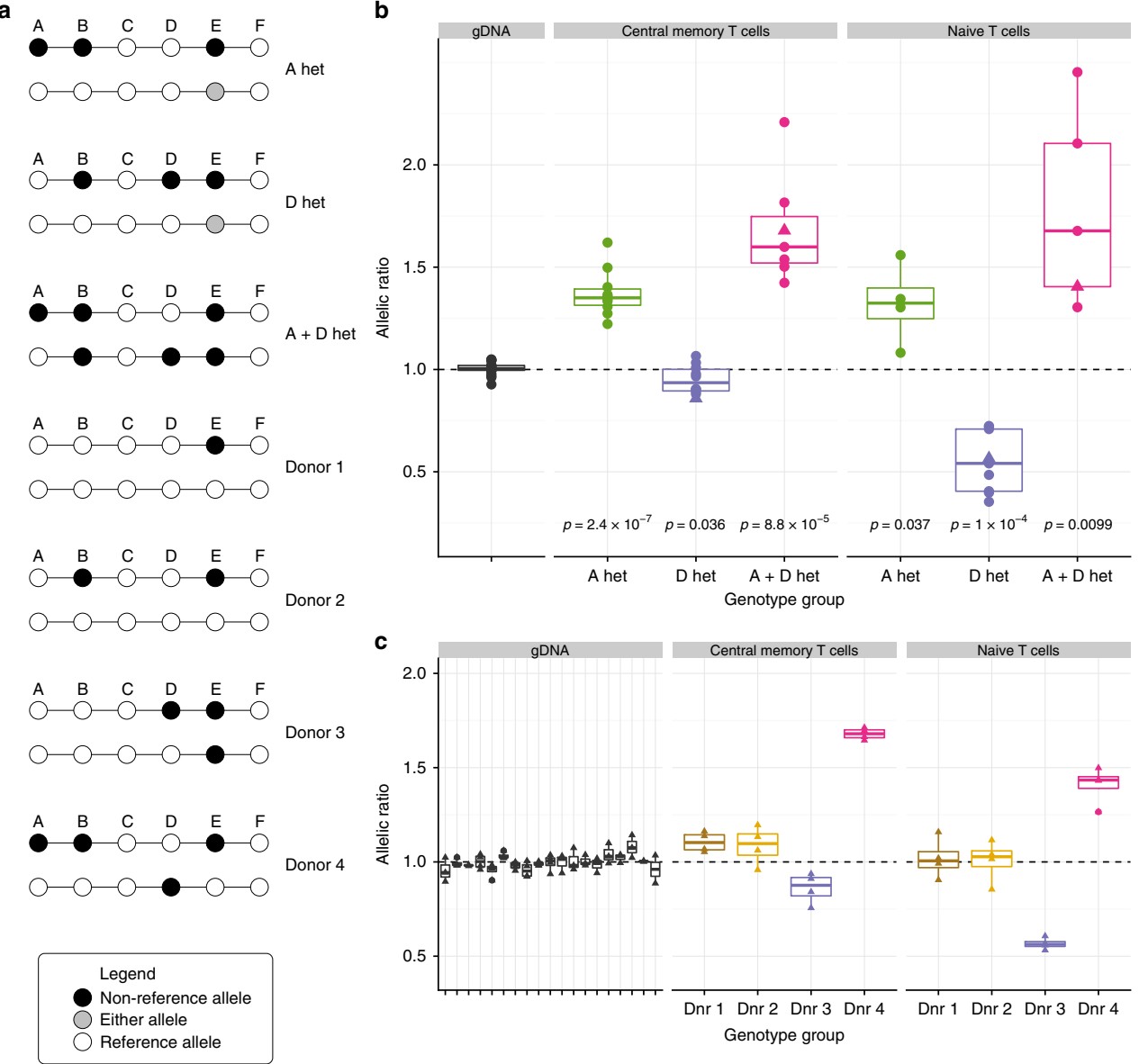

**Fig. 7** Allele-specific expression analysis of *IL2RA*. This shows that there are two phenotypes that map to the A and D SNP groups and not the B group, providing functional evidence that the stochastic search better explains the genetic association than stepwise. **a** Schematic of donor *IL2RA* genotypes used in allele-specific expression studies. As the minor alleles for both A and D each usually co-occur with the minor B allele, in A-het and D-het individuals, the B SNP is heterozygous but in A + D-het individuals, the B SNP is homozygous. There are rare exceptions as seen in donors 3 and 4. **b** Allele-specific expression of *IL2RA* in CD4+ central memory T cells and CD4+ naive T cells in A-het, D-het and A + D het donors. The allelic ratios (top:bottom haplotypes shown in panel a) are calculated from 3–4 replicates per individual. P-values were calculated by *t*-test of allelic ratio in each group compared to gDNA. Donors that contributed to panel c are indicated by a triangle. **c** Donors with rare *IL2RA* haplotypes confirm that the B SNP does not explain genotype-phenotype expression of *IL2RA*. As there is only one donor per genotype we cannot perform statistical testing. We show each of the four replicate allelic ratios (ratio order as in panel **b**) to indicate the variability of the assay. The bounds of each box correspond to the first and third quartiles, and the whiskers extend to the smallest/largest value at most 1.5*IQR from the bounds; IQR is the distance between the first and third quartiles. Data beyond the whisker ends are plotted individually. Source data are provided in Supplementary Data 12

The SNPs belonging to the above-mentioned groups, as well as the lead SNPs for autoimmune thyroid disease (ATD; rs706799), alopecia areata (AA; rs3118470), rheumatoid arthritis (RA; rs10795791) and ulcerative colitis (UC; rs4147359) were extracted from the generated data for analyses via stepwise regression and stochastic search; the lead SNP for multiple sclerosis forms group B. This extraction was done for computational efficiency, and is based on the previous analysis of MS and T1D that identified these SNP groups as contributing the majority of the posterior probability[9]. All other SNPs contribute negligible posterior probability and we assume this in the simulations. The total number of SNPs in the region is not disregarded and is used in the prior probability calculations for the SNPs that are analysed.

For each replication a stepwise regression model was fit, adding SNPs to the model using a p-value threshold of $1 \times 10^{-6}$. To generate stochastic search results, we used GUESSFM[9], setting a prior of three causal variants for the region to encourage good mixing of the chains in the initial Bayesian variable selection, and setting the prior to a more conservative two causal variants per region to obtain final model posterior probabilities (PP). Model fits were summarised by the proportion of times each model was selected via stepwise regression or the mean of the GUESSFM posterior probabilities for each model.

**Simulations—multiple traits**. We adapted the HapGen2 simulation outlined above to simulate datasets for two case and one control set; code is available in

https://github.com/jennasimit/MFMextra. First we used HapGen2 to generate a population of 100,000 individuals based on the CEU 1000 Genomes Phase 3 data. Causal variants for each trait were randomly selected within particular SNP groups for a certain disease model (see Supplementary Data 11); when the same SNP group contained a causal variant for both diseases, one variant was selected from the group and set as causal for both diseases. This reflects our belief that if causal variants for two diseases are known to belong to the same small SNP group, it is likely that the same SNP is causal for both diseases rather than different SNPs in the same high-LD group.

Logistic regression models with the selected causal variants and odds ratios (OR) were then used to assign each individual as either a member of the controls, disease 1 cases, or disease 2 cases until the desired number of individuals in each group was attained; let $OR_{jk}$ be the odds ratio for causal variant $j$ and disease $k$. The prevalence for both diseases was set to 0.1, as our purpose is to generate cases and controls for method comparison. In particular, the following steps were used to ascertain control/disease 1/disease 2 status, where $x_{ij}$ is the number of non-reference alleles of variant $i$ for individual $j$ (i.e. genotype score), $g_j$ is the vector of genotype scores for individual $j$, $\beta_0 = \log(0.1)$, $\beta_{ik} = \log(OR_{ik})$ is the effect of causal variant $i$ for disease $k$, and $m_1$ is the number of causal variants.

1. Let $n_k$ be the number of individuals ascertained to group $k$ (controls are group 0, groups 1 and 2 consist of members with disease 1 and 2, respectively) and $G_k$ be the matrix of genotype scores for individuals in group $k$. Initialise $n_k = 0$ and $G_k$ as a null vector.

2. Set $j = 1$ and repeat the following steps while $n_0 < N_0$ or $n_1 < N_1$ or $n_2 < N_2$.

   a. For $k = 1, 2$ determine $p_{jk} = \text{logit}^{-1}(\beta_0 + \sum_{i=1}^{m_1} \beta_{ik}x_{ij})$ and generate uniform random variables $u_1, u_2$.
   
   b. If $u_1 > p_1$ and $u_1 > p_2$, then $n_0 = n_0 + 1$, append $g_j$ to $G_0$, $j = j + 1$, and go to beginning of step 2.
   
   c. Else, if $u_1 \leq p_1$ and $u_1 > p_2$, then $n_1 = n_1 + 1$, append $g_j$ to $G_1$, $j = j + 1$, and go to beginning of step 2.
   
   d. Else, if $u_1 \leq p_2$ and $u_1 > p_1$, then $n_2 = n_2 + 1$, append $g_j$ to $G_2$, $j = j + 1$, and go to beginning of step 2.
   
   e. Else, if $u_2 < 0.5$ and $n_1 < N_1$, then $n_1 = n_1 + 1$ and append $g_j$ to $G_1$, $j = j + 1$. Otherwise, $n_2 = n_2 + 1$ and append $g_j$ to $G_2$, $j = j + 1$. Go to beginning of step 2.

3. Keep the first $N_k$ rows from $G_k$, $k = 0, 1, 2$.

We simulated either shared configurations where each disease was under the influence of two causal variants, one shared between diseases (A) and one unique to each disease (one from C, one from D); or independent configurations, where the two diseases were under the influence of distinct causal variants (one from each of A and D for one disease and one from C for the other disease) or one disease had no associations in the region (one from each of A and D for one disease and none for the other disease). All causal variants were assigned an odds ratio of 1.25 or 1.4. For both diseases, equal-sized case-control samples consisting of N cases and N controls were considered for N ranging from 1000 to 5000; each simulation setting had 100 replications.

We compared the independent stochastic search analyses of each disease with the multinomial approach with upweighted sharing based on a range of target odds (i.e. prior odds of no sharing of causal variants between one disease and any other disease). We focused on a target odds (TO) of 1, such that there is an equal probability of sharing to non-sharing. Results for a range of TO from 9 (no sharing more likely than sharing of causal variants) to 0.35 (sharing more likely than distinct causal variants) are in Supplementary Data 5–8.

**Mathematical predictions of minimum univariate p-value.** We used sunbeam plots to characterise how changing the odds ratio of two causal SNPs in a model can change the probability that a third variant will have the minimum p-value (and hence be selected first in any stepwise fine-mapping algorithm). We utilised components of the simGWAS package (http://github.com/chr1swallace/simGWAS) to calculate expected GWAS Z scores for any given set of causal variants and their effect sizes, across those causal variants and their neighbouring SNPs[38]. We considered the behaviour of Z scores at each of two nominated causal variants (following Fig. 1, let us refer to these variants as A and C) with a third SNP, not itself causal, but potentially correlated with both A and C (in Fig. 1, this is SNP J). For each of a range of possible odds ratios, we computed which of the three SNPs had the smallest expected p-value, and coloured that square of the grid correspondingly. When the log odds ratios of both A and C were close to 0, then no SNP had a low p-value and it was not possible to find significant evidence of disease association in the region. This section of the grid was coloured white. Superimposed upon the grid is a point corresponding to the odds ratio we computed for A and C from the real dataset. Code to produce these plots is at https://github.com/chr1swallace/MFM-paper/tree/master/sunbeams.

**Fine-mapping analyses of ImmunoChip-genotyped diseases.** We collated individual genotype data generated using the ImmunoChip for a total of 61,641 individuals, formed of controls and six disease cohorts: MS (UK subset)[12], T1D[11], juvenile idiopathic arthritis (JIA, UK subset)[15], celiac disease[14], rheumatoid arthritis

(RA)[16] and autoimmune thyroid disease (ATD)[13] (Supplementary Table 1). All genome coordinates are from build GRCh37.

To ensure controls could be combined across datasets, we restricted analysis for the multinomial model to UK samples, and used principal component analysis including 1000 Genomes data to exclude two individuals who fell outside individual country clusters. Genotypes were compared between datasets to ensure exclusion of duplicate samples. Data were split into subsets according to the densely genotyped regions targeted by the ImmunoChip (Supplementary Data 1) and imputed to 1000 Genomes phase 3[36] using SHAPEIT[39] and IMPUTE2[40]. Phased reference data were downloaded from https://mathgen.stats.ox.ac.uk/impute/1000GP_Phase3.html. Country and the first four principal components were included as covariates in all regressions to account for population structure. SNPs were excluded if they had info scores < 0.3, certainty < 0.98, $|Z|$ for HWE > 4 in UK controls, MAF < 0.5% in UK controls, call rate < 0.99 in any case or control group, or an absolute difference in certain genotype call rates between controls and any case group of >5%.

Forward stepwise regression was performed using univariate logistic regressions across all SNPs in the region. The SNP with the strongest association (smallest p-value) was selected, then all two-SNP models containing the selected SNP and any other SNP were considered, and the process repeated until no SNP could be added with a marginal $p < 10^{-6}$, a less stringent threshold (than the conventional $5 \times 10^{-8}$), chosen so that less than genome-wide significant associations can be considered when other diseases are also associated, in which case borrowing power may be informative.

Stochastic search fine-mapping of single diseases was performed using GUESSFM (http://github.com/chr1swallace/GUESSFM). Initial searches were performed after tagging at $r^2 < 0.99$ with an optimistic binomial prior for the number of causal variants per region with expectation set at 3 to allow good mixing of the chains. Reanalysis of the expanded tag sets for SNPs in models included in the model set with total posterior probability 0.99 was performed using approximate Bayes factors and the more conservative prior expectation of two causal variants per region using GUESSFM. GUESSFM results were combined using the methods proposed in this paper (details in Supplementary Notes 3), as implemented in the R package MFM (http://github.com/jennasimit/MFM). We set the prior odds that two diseases shared any causal variants to 1 (i.e. a 50% probability that they share none). For a number of diseases, $d > 2$, we set the prior that the diseases share no causal variants to $0.5^{\sqrt{d-1}}$, where the exponent is the geometric mean of the exponents in the (nonsensical) extremes $0.5^{d-1}$, which assumes all diseases are independent and 0.5 which assumes all diseases are completely dependent.

Code to perform these steps is available at https://github.com/chr1swallace/MFM-analysis.

**SNP grouping.** SNPs with marginal posterior probability of inclusion >0.001 were grouped according to criteria of substitutability—that is, any single SNP in a group can be used in place of any other to give similar model fits (assessed by Bayes factors) and no more than 1 SNP in the same group is needed to define any models with even modest posterior support. We reasoned that this meant SNPs would need to be in LD—high $r^2$—and rarely selected together in models—i.e. model selection correlation ($r_{model}$) should be negative; both $r_{model}$ and $r^2$ are used so that our SNP grouping is informed by both model posteriors and LD. We hierarchically cluster SNPs within each disease according to $r^2 \times \text{sign}(r_{model})$ using complete linkage, and group SNPs by cutting the tree such that all SNPs within a group must have pairwise $r^2 > 0.5$, pairwise $r_{model} < 0$, and marginal posterior probability that both are included in a model was <0.01. We then identify overlapping groups defined in different diseases, and merge or split groups when they meet this criteria. The specific algorithm is defined in the group.multi function in https://github.com/chr1swallace/GUESSFM/blob/master/R/groups.R.

**Haplotype analyses.** Haplotype analyses were performed by first phasing the genotypes across selected SNPs using an E–M algorithm and selecting 10 multiply imputed samples from the posterior (snphap, https://github.com/chr1swallace/snphap). These samples were analysed in parallel and results combined using standard multiple imputation functions in the R package MICE[41]. Code to implement these steps is available at https://github.com/chr1swallace/snpHaps. All analyses included the first four PCs, and country as an additional covariate for iCEL and iRA to account for population structure.

**Allele-specific expression.** Samples were obtained from the Cambridge BioResource (www.cambridgebioresource.org.uk) as part of the 'Genes and Mechanisms of Type 1 Diabetes' study and were of self-reported white ethnicity. Informed consent was obtained from all volunteers for the collection and use of the peripheral blood samples. The NHS Cambridgeshire Research Ethics committee approved this work involving human participants for allele-specific expression assays. Data and samples were treated anonymously and confidentially.

Allele-specific expression analysis was performed as described in Burren et al., 2017[29] but modified to start with sorted CD4+ naive and central memory T cells. CD4+ naive T cells were sorted as CD3+ CD4+ CD8− CD127med/high CD25low-med

$CD45RA^+$ and $CD27^+$, whereas $CD4^+$ central memory T cells were sorted as $CD3^+$ $CD4^+$ $CD8^-$ $CD127^{med/high}$ $CD25^{low-med}$ $CD45RA^-$ and $CD27^+$.

To phase the direction of effect from the four donors carrying rare IL2RA haplotypes (Fig. 7a, c), their haplotypes were compared to those found in the 1000 Genome Project CEU data to assess the allele frequency of the ASE readout SNP (rs12244380, A or G), to predict which allele is most likely to be carried. For donor 1, the E haplotype carries the G allele with frequency 73% whereas the susceptible haplotype carries the A allele 60% of the time. For donor 2, it is most likely the B and E alleles are on the same haplotype (20 examples where they are together vs four examples where they are on different chromosomes), and here the B + E haplotype carries the A allele of rs12244380 (100%). For donor 3, all examples of the D haplotype lacking the B allele carry the A allele of rs12244380 (14/14), whereas the E haplotype carries the G allele of rs12244380 73% of the time. Lastly, for donor 4, the A haplotype carries the G allele of rs12244380 88% of the time, and for all examples of the D haplotype lacking B carries the A allele of rs12244380 (7/7). Where multiple assays were performed on the same donor, we retained those with the smallest standard deviation of allelic ratios, but show both results in Supplementary Data 12.

## URLs
For Global Biobank Engine, Stanford, CA, see http://gbe.stanford.edu/ [accessed January 2018].

For Extended information in searchable format, see https://chr1swallace.github.io/MFM-output.

**Reporting summary**. Further information on research design is available in the Nature Research Reporting Summary linked to this article.

## Data availability
Complete results from our analyses are available at https://chr1swallace.github.io/MFM-output/index.html and have been deposited at figshare under https://doi.org/10.6084/m9.figshare.8289677. Data were obtained from the study authors for each of the six autoimmune diseases that we analysed. Original genotype data may be requested from the original study authors: ATD ImmunoChip, Cooper et al. (https://www.ncbi.nlm.nih.gov/pubmed/22922229); RA ImmunoChip, Eyre et al. (https://www.ncbi.nlm.nih.gov/pmc/articles/PMC3882906/); JIA ImmunoChip, Hinks et al. (https://www.ncbi.nlm.nih.gov/pubmed/23603761). MS ImmunoChip data was accessed through application to the International Multiple Sclerosis Genetic Consortium (IMSGC; http://www.imsgenetics.org/). Primary analysis of the MS data is presented by IMSGC (https://www.ncbi.nlm.nih.gov/pmc/articles/PMC3832895/). The primary analysis of the Celiac ImmunoChip is by Trynka et al. (https://www.nature.com/articles/ng.998) and the genotype data are hosted by the European Bioinformatics Institute, under accession number EGAS00000000053. T1D ImmunoChip data are available from dbGaP (Study Accession: phs000180.v3.p2) and 2000 T1D samples were genotyped as part of the WTCCC (and controls) - data access is described at https://www.wtccc.org.uk/info/access_to_data_samples.html.

## Code availability
All code used is freely available as R libraries and R scripts. Multinomial Fine-mapping (MFM) software is available at https://jennasimit.github.io/MFM. Custom code for our analyses is available at https://github.com/chr1swallace/MFM-analysis and https://github.com/chr1swallace/MFM-paper. Software for simulations to evaluate MFM is available at https://jennasimit.github.io/MFMextra/.

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

## Acknowledgements

The authors would like to thank the Rivas lab for making the Global Biobank Engine resource available. We gratefully acknowledge the participation of all NIHR Cambridge BioResource volunteers. We thank the Cambridge BioResource staff for their help with volunteer recruitment. We thank members of the Cambridge BioResource SAB and Management Committee for their support of our study and the National Institute for Health Research Cambridge Biomedical Research Centre for funding. Access to Cambridge BioResource volunteers and to their data and samples are governed by the Cambridge BioResource SAB. Documents describing access arrangements and contact details are available at www.cambridgebioresource.org.uk. We thank Graeme Clark, Howard Martin, Fay Rodger and Ruth Littleboy for running the Illumina MiSeq in the Stratified Medicine Core Laboratory NGS hub, Department of Medical Genetics, Cambridge University, Addenbrooke's Hospital, Cambridge. This research was supported by the Cambridge NIHR BRC Cell Phenotyping Hub. In particular, we wish to thank Anna Petrunkina Harrison, Simon McCallum, Christopher Bowman, Natalia Savinykh, Esther Perez and Jelena Markovic Djuric for their advice and support in cell sorting. We also thank Helen Stevens, Pamela Clarke, Gillian Coleman, Sarah Dawson, Jennifer Denesha, Simon Duley, Meeta Maisuria-Armer and Trupti Mistry for acquisition and preparation of samples. We thank Stephen Eyre for helpful comments and disease investigators Stephen Eyre, Wendy Thomson, Gosia Trynka, David van Heel, Steve Rich and the Type 1 Diabetes Genetics Consortium, Stephen Sawcer and the International Multiple Sclerosis Genetics Consortium, Matthew Simmonds, Stephen Gough, Jayne Franklyn, and Oliver Brand for sharing genotype data. This work was supported by the MRC (MC UU 00002/4, MR/R021368/1), the Wellcome Trust (096388, 099772, 107212, 107881), the JDRF (9-2011-253 and 5-SRA-2015-130-A-N), the National Institute for Health Research Cambridge Biomedical Research Centre (BRC), and Dementia Platforms UK. Funding of collection and genotyping of samples: Controls—British 1958 Birth Cohort (MRC grant G0000934, Wellcome Trust (grant 068545/Z/02). DNA control samples were prepared and provided by S. Ring, R. Jones, M. Pembrey, W. McArdle, D. Strachan and P. Burton. UK Blood Services collection of common controls (UKBS-CC collection)—funded by Wellcome Trust (076113/C/04/Z) and by a National Institute for Health Research program grant to National Health Service Blood and Transplant (RP-PG-0310-1002). Genotyping of control samples was supported, in part, by grants from Juvenile Diabetes Research Foundation International (JDRF) and the US NIH (U01 DK062418). ATD—Wellcome Trust Celiac disease—Wellcome Trust, by grants from the Celiac Disease Consortium and an Innovative Cluster approved by the Netherlands Genomics Initiative, by the Dutch Government (BSIK03009 to C. Wijmenga) and the Netherlands Organisation for Scientific Research (NWO, grant 918.66.620) and by the US National Institutes of Health grant 1R01CA141743 and Fondo de Investigacin Sanitaria grants FIS08/1676 and FIS07/0353. JIA—Arthritis Research UK (grant 17552). Sparks Childhood Arthritis Response to Medication Study was funded by Sparks, UK (08ICH09) and the Big Lottery Fund, UK (RG/1/010135231). MS—US National Institutes of Health, the Wellcome Trust, the UK MS Society, the UK Medical Research Council, the US National MS Society, the Cambridge National Institute for Health Research (NIHR) Biomedical Research Centre, DeNDRon, the Bibbi and Niels Jensens Foundation, the Swedish Brain Foundation, the Swedish Research Council, the Knut and Alice Wallenberg Foundation, the Swedish Heart–Lung Foundation, the Foundation for Strategic Research, the Stockholm County Council, Karolinska Institutet, INSERM, Fondation d'Aide pour la Recherche sur la Sclrose en Plaques, Association Franaise contre les Myopathies, Infrastrutures en Biologie Sant et Agronomie (GIS–IBISA), the German Ministry for Education and Research, the German Competence Network MS, Deutsche Forschungsgemeinschaft, Munich Biotec Cluster M4, the Fidelity Biosciences Research Initiative, Research Foundation Flanders, Research Fund KU Leuven, the Belgian Charcot Foundation, Gemeinntzige Hertie Stiftung, University Zurich, the Danish MS Society, the Danish Council for Strategic Research, the Academy of Finland, the Sigrid Juselius Foundation, Helsinki University, the Italian MS Foundation, Fondazione Cariplo, the Italian Ministry of University and Research, the Torino Savings Bank Foundation, the Italian Ministry of Health, the Italian Institute of Experimental Neurology, the MS Association of Oslo, the Norwegian Research Council, the South–Eastern Norwegian Health Authorities, the Australian National Health and Medical Research Council, the Dutch MS Foundation and Kaiser Permanente. RA—Arthritis Foundation, the US National Institutes of Health, and Arthritis Research UK (21754). T1D—National Institute of Diabetes and Digestive and Kidney Diseases (NIDDK), National Institute of Allergy and Infectious Diseases (NIAID), National Human Genome Research Institute (NHGRI), National Institute of Child Health and Human Development (NICHD), and the JDRF and supported by U01 DK062418.

## Author contributions

J.A. developed the MFM method, performed statistical analyses and interpreted results, wrote the paper. D.R. performed ASE assays and analysis and interpreted results, wrote the paper. M.F. performed mathematical analyses and conceived the sunbeam plots. N.G. performed mathematical analyses and verified the analytical methods. L.W. supervised ASE work, interpreted the results of ASE analyses and all the fine-mapping analyses, wrote the paper. C.W. conceived the study, developed the MFM method, performed statistical analyses and interpreted results, wrote the paper. All authors read and agreed the manuscript.

## Additional information

**Competing interests:** The authors declare no competing interests.

