## [Peer Review File · Nature Communications]

Reviewers' Comments:

Reviewer #1:

Remarks to the Author:

The authors propose a novel approach to fine-map genetic variants that might be missed by more traditional statistical approaches by using multinomial regression ideas to simultaneously fine-map genetic variants for multiple diseases, where the disease are likely to share genetic causes.

Intuitively, the ideas make good sense. Furthermore, the motivating of haplotypes in in Figure 1 nicely illustrates the complexity that might occur when haplotypes have complex patterns of association with diseases. I also like the descriptions of how stepwise selection can get "stuck" on the tag, showing why stochastic search can be better because it considers the joint effects of SNPs. Another strength of this paper is the large and rich data sets used for analysis of six immunologic diseases.

However, I found the paper difficult to follow. Early in the results section, the authors call SNP rs706779 as "group J", and in the following paragraph discuss the "J model", but this model is not defined (Figure 1 gives some hints that are insufficient). The text also discusses "A+C" model, which is not clearly defined. This pattern of discussing undefined models throughout the text make the paper difficult to understand. For example, the results discuss models "I+K" and "E+G", but it is not clear what these represent. Do these represent configurations of casual variants, or patterns of haplotypes (as given in the example of Figure 1)? It is also stated that SNPs in high-LD are grouped together, so it is not clear how these are denoted by model notation.

I also found the text to be ambiguous regarding the haplotypes discussed in Figure 1. The text discusses the order of SNPs as rs706..., rs618..., and rs115..., but Figure 1 has the order as rs618..., rs706..., and rs115.... Also, the text says the common haplotype is TCT but Figure 1 has the common haplotype as CTT. Furthermore, the colored lines in Figure 1 have no legend to make it clear what the color indicates.

Lines 126-127 suggest that 20-40% of potential common variant pairs have a potential joint tag. This seems pretty speculative, based on unverified assumptions.

What do the yellow/red colors in Figure 2 represent? Are the D/E regions for extreme correlations that have the same signs? It is stated that regions B/C are for when correlations have opposite signs, but what about the regions where correlations have the same sign, but are in the yellow areas?

On lines 177-178 it is stated that kappa is chosen when you know the prior probability, but in real data, how is kappa chosen? Is it always assumed to be 0.5?

For Figure 5, should there be a negative sign in front of $\log(\text{MAF})$?

For stepwise regression, the p-value threshold of $10E-6$ was used for model selection. How was this threshold chosen? Would the performance of stepwise differ much if you chose a different threshold? I find it difficult to interpret the comparisons of the stepwise selection with stochastic search and the Bayes model when only a single threshold is used for comparisons. For fine-mapping, it seems to me that you are already convinced that a region harbors a causal variant, so conditional on this, wouldn't you want to use a less stringent threshold?

For SNP grouping, SNPs had to have $r^2 > 0.5$ and $r < 0$, based on the reasoning that the SNPs should be rarely selected together. However, it is possible for SNPs to not be selected together if r is large, say $r > 0.9$ or $r > 0.95$, because if one SNP enters the model, the second SNP explains very little, conditional on the first SNP. So, I don't understand why you only required $r < 0$.

Reviewer #2:

Remarks to the Author:

This was overall a fairly interesting manuscript that (a) explores the benefits of stochastic search over stepwise regression for fine mapping loci in genetic association studies and (b) proposes a new method (MFM) to exploit the results of such analyses carried out for multiple traits in order to improve power for fine-mapping any given trait. A reasonably compelling functional validation (via allelic-specific expression assays) of the results from fine-mapping one such locus (IL12RA) is presented.

I have a number of comments, some minor and some more substantive.

General comments:

1. The first part of the manuscript (pages 3-9 and 8 pages of the Suppl Note, sections 1 and 2) focuses heavily on the exploring benefits of stochastic search over stepwise regression for fine mapping loci, before moving on to describing/applying the new MFM method. This emphasis is not, however, reflected in the title which suggests that the paper is "all about" the new MFM method. I suggest adjusting the title to better reflect the overall focus of the manuscript

2. The whole issue of stochastic search offering a benefit over stepwise regression is actually not that novel, although this manuscript does provide a nice demonstration of it. But the general principle has already been demonstrated in previous papers by the authors and others (and is indeed anyway pretty "obvious" from first principles). Actually, much of the benefit demonstrated (related to LD patterns and tagging on two causal variants by a single variant) would be equally applicable to exhaustive search as stochastic search - presumably the main benefit of stochastic search is that it "scales up" better - although for fine-mapping within a locus (as here) I would have thought that exhaustive search of all two-way, all 3-way, possibly all 4-way combinations would not be infeasible. Some further discussion of all this would be helpful.

3. The method has some conceptual similarities with the recently proposed moloc method (Giambartolomi et al. 2018 *Bioinformatics* 34(25) 3538-2545), although I think the emphasis is probably a bit different - the moloc method appears to focus on identifying which set of traits co-localise, whereas MFM rather exploits the co-localisation of several traits to fine-map for each individual trait (which would presumably identify which set of traits co-localise as a side-product). It would be helpful to provide some discussion of the similarities and differences between MFM and moloc.

4. The actual fine-mapping method used throughout would seem to be GUESSFM, developed by the authors. There are by now a number of fine-mapping methods around (e.g. JAM, FINEMAP) which do not necessarily give concordant results - see for example Darlay et al. 2018 *PLOS Genetics* 14(12) e1007833. While not expecting the authors to re-run all of their analyses using several different methods, I think it would be worth exploring, at least for the single-disease analyses, how different the results are when using JAM and/or FINEMAP compared to GUESSFM. Also providing some discussion of how easily MFM could be used with the output of one of these other fine-mapping methods instead of GUESSFM, if desired. (Which might be attractive since I believe GUESSFM requires individual level data (?) while JAM and FINEMAP require only summary statistics).

Specific comments:

5. Abstract lines 4-5: "... can mis-identify as causal, SNPs which jointly tag distinct causal variants". I found this sentence a bit confusing, especially regarding what exactly was meant by "jointly" (it suggests that 2 SNPs might jointly tag a single causal variant - which I think is exactly

the opposite phenomenon from what is being intended!) How about "... can mis-identify as causal, a SNP that tags several distinct causal variants"?

6. Page 3 line 25: "The problem is often approached through stepwise regression [4]" - it is true that the GCTA implementation referenced (Yang et al. 2012) is often used on account of its computational convenience and applicability to summary statistics. However, I think the first use of this type of stepwise regression approach for genetic fine mapping within a locus is probably Cordell and Clayton 2002 Am J Hum Genet 70(1) 124-141. Might be worth referencing both papers.

7. Page 13 line 204: "When the multinomial is inappropriate for all samples" - does this include the situation where there are in fact no shared controls? (As might occur for analyses based on summary statistics, rather than individual level data)?

8. Page 15 Figure 4: I found this figure quite hard to navigate. Would it be clearer to use the subtitles "Disease 1" and "Disease 2" (as done in panel a) throughout? For panel b it looks to me as if Disease 1 and Disease 2 have actually been swapped around (Disease 2 on the left, Disease 1 on the right)?

9. Page 28 line 503: "The SNPs belonging to the above-mentioned groups" - why not use all 345 SNPs? Is that too computationally intensive? But then that makes the simulation not particularly realistic...

10. Page 29 lines 518-521. Does this mean you set one variant (the same variant) in a group as causal for both diseases? What is the rationale for doing this rather than keeping the different causal SNPs (within the same group) chosen for the different diseases?

11. Page 38-40 (Tables 1 and 2) - while appreciating the rationale for comparing the "top" models from stepwise and stochastic search, or from stochastic search and MFM, actually this seems a bit of a simplistic comparison. One of the attractive features of stochastic search/MFM is that one does not have to choose the "top" model - one rather ends up with a set of competing models, each with their posterior probabilities.

Minor comments:

Page 34 line 605-606: "using the methods proposed in this paper" - I think you need to refer people to the Supplementary Note here, since that would appear to be the place where the MFM method is actually described.

Page 40-42 (Tables 2-4): There seems to be a disconnect between the figure legends and the table column headings. In the figure legends you use the descriptors "stochastic search" and "MFM" while in the Table column headings you seem to be using the terms "Independent" and "Joint". It would be helpful to use the same terms in the Table column headings as are used in the legends.

Supplementary Note Page 1: you seem to be missing several members of the authorship?!

Supplementary Note Page 10: "have been derived [?]" - missing reference?

Legend to Suppl Fig 3: "MMPI > 0.5". (i.e. use > as opposed to the strange upside down ? sign).

Response to Referees for “Stochastic search and joint fine-mapping increases accuracy and identifies novel associations in six immune-mediated diseases”

Reviewer #1 (Remarks to the Author):

The authors propose a novel approach to fine-map genetic variants that might be missed by more traditional statistical approaches by using multinomial regression ideas to simultaneously fine-map genetic variants for multiple diseases, where the disease are likely to share genetic causes. Intuitively, the ideas make good sense. Furthermore, the motivating of haplotypes in in Figure 1 nicely illustrates the complexity that might occur when haplotypes have complex patterns of association with diseases. I also like the descriptions of how stepwise selection can get “stuck” on the tag, showing why stochastic search can be better because it considers the joint effects of SNPs. Another strength of this paper is the large and rich data sets used for analysis of six immunologic diseases.

However, I found the paper difficult to follow. Early in the results section, the authors call SNP rs706779 as “group J”, and in the following paragraph discuss the “J model”, but this model is not defined (Figure 1 gives some hints that are insufficient). The text also discusses “A+C” model, which is not clearly defined. This pattern of discussing undefined models throughout the text make the paper difficult to understand. For example, the results discuss models “I+K” and “E+G”, but it is not clear what these represent. Do these represent configurations of casual variants, or patterns of haplotypes (as given in the example of Figure 1)? It is also stated that SNPs in high-LD are grouped together, so it is not clear how these are denoted by model notation.

We recognise that presenting results via SNP groups rather than individual SNPs is uncommon, but decided that this was the best way to summarise inference across the thousands of models considered for each trait. Each group is defined as a set of SNPs in LD, that also meet our definition of "substitutability". We have now clarified this definition of substitutability under the heading "SNP grouping" in the Methods:

“that is, any single SNP in a group can be used in place of any other to give similar model fits (assessed by Bayes factors) and no more than 1 SNP in the same group is needed to define any models with even modest posterior support”

When we talk about the "A+C model" this means we are talking about the collection of models that contain exactly one SNP from group A and one from group C and no others. We have now clarified this in the second paragraph of results, and highlight there also that the membership of each SNP group is given in Supplementary Table 4:

“When we discuss a SNP group model, e.g. “model A+B”, we mean the collection of models that include exactly one SNP from group A and exactly one SNP from group B, and no others. We prefer to consider posterior support for each grouped model (“gPP”) as the sum of posterior probabilities over all SNP models in that group when interpreting

the stochastic search results. SNP group membership is shown in Supplementary Table 4."

When presenting haplotypes, the high LD within SNP groups means showing more than one SNP from the same group is often redundant, and so we show a single representative group member. We emphasize this now by stating "A representative SNP from each SNP group is shown." in the legend of the first haplotype plot.

For clarity, in the Results Section, Paragraph 3, we changed "treating SNPs in strong LD as equivalent" to "treating SNPs in the same SNP group as equivalent"

Within the first paragraph of the "Joint tagging of stochastic search models by stepwise SNPs" section we added "a member of" and "from group" qualifiers to the first instances of J, A, and C.

In the 2nd paragraph of "Joint tagging of stochastic search models by stepwise SNPs" we clarified the first instances of the J and A+C models:

"J model (any model with exactly one SNP from group J)"

"A+C model (2-SNP model with a SNP from each of groups A and C)"

and where to find SNP group details:

"SNP group membership in Supplementary Table 4"

In the "MFM analysis of up to six IMD" section, we clarified the E+H notation:

"...while RA and T1D both have 2 signals, in groups labelled E and H, represented by causal variant configuration E+H.

and where to find SNP group details:

"SNP group membership in Supplementary Table 4"

In the Figure 1 legend we add the clarification that "A representative SNP from each SNP group is shown."

I also found the text to be ambiguous regarding the haplotypes discussed in Figure 1. The text discusses the order of SNPs as rs706..., rs618..., and rs115..., but Figure 1 has the order as rs618..., rs706..., and rs115.... Also, the text says the common haplotype is TCT but Figure 1 has the common haplotype as CTT. Furthermore, the colored lines in Figure 1 have no legend to make it clear what the color indicates.

Thank you for flagging this mismatch. In the first paragraph of the "Joint tagging of stochastic search models by stepwise SNPs" section, we have changed the ordering of SNPs to match the order given in Figure 1a: rs61839660/A, rs706779/J and rs11594656/C.

The haplotypes were listed by the SNP ordering in the text, rather than figure, and we have corrected this; CTT is the common haplotype and TCT or CCA are protective for ATD.

The legend of Figure 1 now includes a colour legend for the models: "Potential models include J (green), C (blue), A+C (purple), A (red) and null (black); any other models are grouped together as grey."

Lines 126-127 suggest that 20-40% of potential common variant pairs have a potential joint tag. This seems pretty speculative, based on unverified assumptions.

We assumed equal odds ratios for the causal variants, and for this reason the 20-40% is an upper bound. At the end of the 4th paragraph in the "Joint tagging of stochastic search models by stepwise SNPs" section, we have clarified this:

"Doing so, we found that 20-40% of potential common causal variant pairs (MAF>5%) had a potential joint tag, though this was highly variable across regions (Fig 2b-c) and should be considered an upper limit because effect sizes may not be equal at neighbouring causal variants."

What do the yellow/red colors in Figure 2 represent? Are the D/E regions for extreme correlations that have the same signs? It is stated that regions B/C are for when correlations have opposite signs, but what about the regions where correlations have the same sign, but are in the yellow areas?

The take home message from this figure should be that
a *only when a third SNP is strongly correlated with both causal variants, in the same direction (such that the effects add up rather than cancel each other out) can joint tagging occur*
b *even assuming the most favourable situation for this to happen (equal effect sizes) it is rare for LD patterns to enable tagging for any randomly chosen trio of SNPs, and*
c *because there are many possible third SNPs for any pair of causal variants, the chance that at least one SNP could tag two causal variants is considerably higher than we might expect given the low probability of tagging-compatible LD in any individual trio*

We have clarified the Figure 2 legend with regards the meaning of red/yellow and D/E as follows:

Figure 2: Potential frequency of joint tagging. We consider the patterns of three-way LD between each possible trio of SNPs, nominating the first two as causal, and the third as a potential tag. **a** For each pair of potential causal SNPs, we can predict whether the third SNP is a tag according to the pairwise correlation between that SNP and the two potentially causal SNPs (r_1 , r_2). Red (yellow) areas indicate settings where the third SNP is (is not) a potential tag for SNPs 1 and 2. In this example, the potentially causal SNPs have equal MAF, equal effect on disease risk (equal odds ratios, OR) and are uncorrelated. Then, if the third SNP is (A) uncorrelated or weakly correlated with either SNP 1 or 2, or (B, C) negatively correlated with one and positively with the other, we would not expect it to act as a tag. On the other hand, if it were (D, E) strongly positively or negatively correlated with both causal variants, we would expect it to act as a joint tag. **b** shows the result of searching all possible SNP trios in UK ImmunoChip control data, and quantifying the proportion of trios that correspond to joint tagging

in each region, assuming the causal variants have equal OR; the pattern is individually rare, consistently <5%. **c** shows the proportion of SNP pairs for which at least one potential tag exists, which can be substantial - about 40% overall.

On lines 177-178 it is stated that kappa is chosen when you know the prior probability, but in real data, how is kappa chosen? Is it always assumed to be 0.5?

We have added clarification that this value was chosen as “compatible with conclusions of previous studies of IMDs”. The 0.5 prior was chosen based on prior work and otherwise a range of plausible values should be considered. We have added a discussion on this to the end of the 2nd paragraph in the Discussion:

"In the case of other collections of diseases, researchers will need to elicit appropriate priors for each collection. This might be possible from prior work as here, but if not we recommend that a range of plausible values be considered, with robust results identified as those which remain similar under different priors."

For Figure 5, should there be a negative sign in front of log(MAF)?

No, but thank you for noticing - we should have stated that MAF is shown as a percentage, on a log scale to allow frequencies of rarer haplotypes to be distinguished, which we have now added to the Figure legend.

For stepwise regression, the p-value threshold of $10E-6$ was used for model selection. How was this threshold chosen? Would the performance of stepwise differ much if you chose a different threshold? I find it difficult to interpret the comparisons of the stepwise selection with stochastic search and the Bayes model when only a single threshold is used for comparisons. For fine-mapping, it seems to me that you are already convinced that a region harbors a causal variant, so conditional on this, wouldn't you want to use a less stringent threshold?

We agree that, conditional on seeing a genome-wide significant SNP, we may be more ready to believe association at a second SNP at a less stringent threshold. However, our experience is that there will be insufficient information to fine map that second signal unless it also shows reasonably strong association. We do consider $10E-6$ to be a less stringent threshold (than the conventional $5E-8$), chosen so that less than genome-wide significant associations can be considered, while balancing the need for informative fine mapping. A note on this choice was added to the last paragraph of the "Fine mapping analyses of ImmunoChip-genotyped diseases" section in Methods:

" 1×10^{-6} , a less stringent threshold (than the conventional 5×10^{-8}), chosen so that less

than genome-wide significant associations can be considered when other diseases are also associated, in which case borrowing power may be informative.”

Comparison between Bayesian and stepwise search results are complicated not only by the use of a hard threshold in stepwise search, but also because of the ordering of SNPs.

For SNP grouping, SNPs had to have $r^2 > 0.5$ and $r < 0$, based on the reasoning that the SNPs should be rarely selected together. However, it is possible for SNPs to not be selected together if r is large, say $r > 0.9$ or $r > 0.95$, because if one SNP enters the model, the second SNP explains very little, conditional on the first SNP. So, I don't understand why you only required $r < 0$.

Our terminology here appears to be confusing. r^2 refers to the population level correlation between SNP genotypes. r refers to the correlation between SNPs according to whether they are selected together in the models with posterior support, thus $r < 0$ indicates a tendency for SNPs not to be selected together. We have replaced r by r_{model} to better distinguish the two sources of correlation used. We have also clarified the text in this section, adding:

“both r_{model} and r^2 are used so that our SNP grouping is informed by both model posteriors and LD”.

Reviewer #2 (Remarks to the Author):

This was overall a fairly interesting manuscript that (a) explores the benefits of stochastic search over stepwise regression for fine mapping loci in genetic association studies and (b) proposes a new method (MFM) to exploit the results of such analyses carried out for multiple traits in order to improve power for fine-mapping any given trait. A reasonably compelling functional validation (via allelic-specific expression assays) of the results from fine-mapping one such locus (IL12RA) is presented.

I have a number of comments, some minor and some more substantive.

General comments:

1. The first part of the manuscript (pages 3-9 and 8 pages of the Suppl Note, sections 1 and 2) focuses heavily on the exploring benefits of stochastic search over stepwise regression for fine mapping loci, before moving on to describing/applying the new MFM method. This emphasis is not, however, reflected in the title which suggests that the paper is “all about” the new MFM method. I suggest adjusting the title to better reflect the overall focus of the manuscript

We have updated the abstract to better reflect the focus of the manuscript and have altered the title to

“Stochastic search and joint fine-mapping increases accuracy and identifies novel associations in six immune-mediated diseases”.

The abstract now includes:

“Both lack of power, and “joint tagging” of two or more distinct causal variants by a single non-causal SNP, lead to inaccuracies in fine-mapping, with stochastic search more robust than stepwise.”

2. The whole issue of stochastic search offering a benefit over stepwise regression is actually not that novel, although this manuscript does provide a nice demonstration of it. But the general principle has already been demonstrated in previous papers by the authors and others (and is indeed anyway pretty “obvious” from first principles). Actually, much of the benefit demonstrated (related to LD patterns and tagging on two causal variants by a single variant) would be equally applicable to exhaustive search as stochastic search - presumably the main benefit of stochastic search is that it “scales up” better - although for fine-mapping within a locus (as here) I would have thought that exhaustive search of all two-way, all 3-way, possibly all 4-way combinations would not be infeasible. Some further discussion of all this would be helpful.

We are glad the reviewer sees that the benefits of a non-stepwise search are pretty obvious. While not new, we do deliberately labour this point because our discussions with others in the GWAS field have shown us that many people are unconcerned.

We do indeed use stochastic search rather than exhaustive search indeed for the scalability. We have now added the following text to the first paragraph of the discussion:

“In principle, exhaustive search could be used to overcome the problem, but scalability is a substantial problem: in a 1000-SNP region: there are a manageable 0.5 million 2-way models but over 166 million 3-way models (and 41.5 billion 4-way models) which cannot be fit in reasonable time for a logistic model that requires optimisation for each model. This exponential growth of model numbers has motivated different approaches to scalable search strategies using linear models and exhaustive search(Chen et al. 2015; Guan and Stephens 2008) or specialised search strategies(Benner et al. 2016; Newcombe et al. 2016; Bottolo et al. 2013). We show that in contrast, a logistic model stochastic search(Wallace et al. 2015) is feasible and does tend to the correct solution as sample sizes increase”

3. The method has some conceptual similarities with the recently proposed moloc method (Giambartolomi et al. 2018 *Bioinformatics* 34(25) 3538-2545), although I think the emphasis is probably a bit different - the moloc method appears to focus on identifying which set of traits co-localise, whereas MFM rather exploits the co-localisation of several traits to fine-map for each individual trait (which would presumably identify which set of traits co-localise as a side-product). It would be helpful to provide some discussion of the similarities and differences between MFM and moloc.

We have added moloc to its predecessor (coloc) in the second paragraph of our Discussion and extended the discussion to compare the two approaches:

“Our new method MFM borrows information across diseases and is thus related to, but distinct from, methods that compare two(Giambartolomei et al. 2014) or more(Giambartolomei et al. 2018) traits which integrate over the fine mapping posteriors of individual traits, upweighting models that share causal variants, to determine whether there is evidence for sharing. Here, we exploit a prior belief that traits studied are enriched for colocalisation to determine the marginal fine mapping posterior for each trait, and remove the common colocalisation assumptions of independent datasets and a single causal variant per trait in any region.”

4. The actual fine-mapping method used throughout would seem to be GUESSFM, developed by the authors. There are by now a number of fine-mapping methods around (e.g. JAM, FINEMAP) which do not necessarily give concordant results - see for example Darlay et al. 2018 PLOS Genetics 14(12) e1007833. While not expecting the authors to re-run all of their analyses using several different methods, I think it would be worth exploring, at least for the single-disease analyses, how different the results are when using JAM and/or FINEMAP compared to GUESSFM. Also providing some discussion of how easily MFM could be used with the output of one of these other fine-mapping methods instead of GUESSFM, if desired. (Which might be attractive since I believe GUESSFM requires individual level data (?) while JAM and FINEMAP require only summary statistics).

All Bayesian results are a product of the prior structure, statistical model of the data used, and in the case of fine mapping, the search strategy used. We chose GUESSFM because it is the only stochastic search method specifically adapted to a logistic regression model and thus appropriate for both the binary outcome (disease/no disease) and the retrospective sampling design, while the other methods use a linear statistical model.

GUESSFM does indeed require individual level data, but this is necessary for reanalysis under a multinomial model that requires controls to be shared between studies. We agree MFM could be used with the output of any of these other methods if datasets were independent or controls completely shared, and we have added a sentence to that effect to the end of the first paragraph of the Discussion:

“MFM could be easily adapted as a wrapper around any of the linear model methods above, with the conditions that the linear model is considered an acceptable approximation to a logistic model and that controls would need to be either shared completely or not at all.”

We have not yet done the programming work to enable this, but that is certainly feasible and planned as part of a current collaborative project.

While we used GUESSFM for theoretical and practical reasons, we have now explored the difference between results running GUESSFM, JAM or FINEMAP, and set these out below. We

would rather not include these in the manuscript because we consider the main message of the paper to be that stochastic search, and combination of data across diseases, are necessary to find the best fitting disease models. Detailing the differences between GUESSFM and FINEMAP would distract from this - FINEMAP is a strong stochastic search method particularly suited for quantitative traits, but in most cases cannot be adapted to approximate a multinomial likelihood across multiple studies simply because most studies do not completely share controls, but variably share different subsets of them, and so it is necessary to have access to the control genotype data.

We ran JAM and FINEMAP on all the models highlighted as giving different stepwise/stochastic search model sizes in Tables 1-2. JAM often failed to converge, but when it did mostly found the same model as stepwise, which we consider is likely due to the simple add/remove moves in the MCMC algorithm which may get stuck in a local mode. FINEMAP uses a stochastic shotgun search that is therefore more comparable to GUESSFM, and we found the top GUESSFM model was always amongst the top FINEMAP models, but that the FINEMAP posterior was generally flatter over the different models, with few models standing out as strongly supported. We think this is due partly to the need to thin input SNPs to remove those in very high LD for FINEMAP (which is not needed for GUESSFM), due to different priors on the effect sizes, and due to using a prospective linear model for a data sampled retrospectively on a binary outcome.

The results of FINEMAP and GUESSFM are compared below. Using the SNP groups identified by GUESSFM, we display the group posterior probability (GPP) for both methods. The model selected by GUESSFM is displayed in bold for FINEMAP for ease of comparison, and for this model we also give the size of each SNP group (Total) and the number of SNPs that are used in FINEMAP (Used - retained after filtered with $r^2=0.99$); FINEMAP requires that all off-diagonal elements in the LD matrix are between -1 and 1 so that filtering of SNPs is needed before use. GUESSFM uses tag SNPs and then expands the models, such that all filtered SNPs are substituted back into the models according to their tag. This step is not used by FINEMAP and is likely the cause of any discrepancies in top model since the GPP is based on a subset of the SNPs.

Region, Disease	GUESSFM			FINEMAP		
	Top 2 Models	GPP	logBF	Best Models (GPP > 0.01)	GPP	logBF
2q-204446380-204816382, T1D	E+H	0.765	41.04	E+H+L	0.045	20.80
	E+H+J	0.060	44.67	E+H Total: E:31, H:52 Used: E:7, H:12	0.038	18.49

				E+H+L+rs76562827	0.026	22.92
				E+H+rs76562827	0.018	22.75
				E+H+J	0.012	20.07
				E+H+L+rs4335928	0.012	22.80
4q-122973062-123565302, T1D	A+F	0.85	29.74	A+F Total: A:106, F:53 Used: A:4, F:3	0.032	13.31
	D	0.019	23.62	A+F+rs78956800	0.019	15.17
				B+D	0.012	13.36
				D+rs114088632	0.012	13.45
10p-6030000-6220000, ATD	A+C	0.954	19.04	A+C Total: A: 31, C: 8 Used: A: 11, C: 6	0.313	9.16
	J	0.011	10.18	A+C+X10	0.028	9.98
				A+C+rs112613053	0.021	10.01
				A+C+rs62626310	0.017	10.18
14q-101290463-101328739, T1D	A+B	0.777	19.48	A+B Total: A: 16, B: 5 Used: A: 4, B: 4	0.290	10.37
	C	0.081	13.59	A+B+rs45617834	0.042	11.23
				A+B+rs3742390	0.026	10.98
				A+B+rs10147988	0.022	10.91

				A+B+rs115074173	0.020	10.90
				B+C	0.014	8.68
				A+B+rs2400941	0.011	10.60
10p-6030000-6220000, MS	B	0.632	23.18	A+D	0.105	12.45
	A+D	0.188 (MFM; 0.833)	26.41	B+H	0.049	12.35
				B Total: 3 Used: 2	0.041	10.15
				B+D	0.019	12.04
				A+D+rs79582739	0.013	13.87
				A+D+rs12722497	0.013	13.73
				A+D+rs7078273	0.012	13.69
				A+D+H	0.010	13.13
				A+B+D	0.010	13.40
16p-11017058-11307024, MS	A	0.381	18.04	B+D Total: B: 89, D: 17 Used: B: 16, D: 5	0.141	11.20
	B+D	0.175 (MFM; 0.347)	21.34	A+B	0.054	10.62
				B+D+rs79263553	0.046	13.04

				B+D+rs117714844	0.044	13.03
				A+B+D	0.036	12.06
				B+D+G	0.028	12.20
				A+D	0.024	10.60
				B+F	0.019	10.42
				A+B+rs79263553	0.015	12.21
				B+rs79263553	0.014	10.66
				B+G+rs79263553	0.013	12.22
				B+G	0.011	9.82

Specific comments:

5. Abstract lines 4-5: "... can mis-identify as causal, SNPs which jointly tag distinct causal variants". I found this sentence a bit confusing, especially regarding what exactly was meant by "jointly" (it suggests that 2 SNPs might jointly tag a single causal variant - which I think is exactly the opposite phenomenon from what is being intended!) How about "... can mis-identify as causal, a SNP that tags several distinct causal variants"?

Thank you for noticing the need for clarification of "jointly" and we have made the change to "Both lack of power, and "joint tagging" of two or more distinct causal variants by a single non-causal SNP, lead to inaccuracies in fine-mapping, with stochastic search more robust than stepwise. "

6. Page 3 line 25: "The problem is often approached through stepwise regression [4]" - it is true that the GCTA implementation referenced (Yang et al. 2012) is often used on account of its computational convenience and applicability to summary statistics. However, I think the first use of this type of stepwise regression approach for genetic fine mapping within a locus is probably Cordell and Clayton 2002 Am J Hum Genet 70(1) 124-141. Might be worth referencing both papers.

Thank you for reminding us of this reference and we have now added it in, referencing both papers (2nd paragraph of Introduction).

7. Page 13 line 204: "When the multinomial is inappropriate for all samples" - does this include the situation where there are in fact no shared controls? (As might occur for analyses based on summary statistics, rather than individual level data)?

This is true and has been added in at the end of the third paragraph of the "Proposed method for multinomial fine mapping (MFM) of multiple diseases" section:

"A multinomial model accounts for shared controls across diseases; when controls are not shared, the joint log Bayes factor is a simple sum of logistic log Bayes factors. This allows us to deal with multiple populations, with not all populations represented for all diseases, where we fit a multinomial to the samples from common populations with shared controls, and add disease-specific log Bayes factor terms from logistic models fitted to the distinct populations."

8. Page 15 Figure 4: I found this figure quite hard to navigate. Would it be clearer to use the sub-titles "Disease 1" and "Disease 2" (as done in panel a) throughout? For panel b it looks to me as if Disease 1 and Disease 2 have actually been swapped around (Disease 2 on the left, Disease 1 on the right)?

This figure does contain lots of information and we have made it again, using the sub-titles "Disease 1" and "Disease 2" for all panels, and listing the simulation details in the legend, which has been re-written for clarity:

*"Comparison of MFM analysis and single disease analysis. Causal variants were simulated for two diseases with models defined by SNP groups from the IL2RA region. MFM is shown by solid lines and independent analyses by dashed lines. Throughout, disease 1 has causal variants A+D, while causal variants for disease 2 vary. **a,b** Disease 2 has causal variants A+C and the odds ratio of A, OR_A , is the same for both diseases; **a** A has a stronger effect than C and D; $OR_A = 1.4$ (both), $OR_D = 1.25$ (disease 1), $OR_C = 1.25$ (disease 2). **b** A has a weaker effect than C and D; $OR_A = 1.25$ (both), $OR_D = 1.4$ (disease 1), $OR_C = 1.4$ (disease 2). **c** Disease 2 has only C causal; $OR_A = 1.25$ (both), $OR_D = 1.25$ (disease 1), $OR_C = 1.25$ (disease 2). **d** Disease 2 has no causal variants (no association). **a,b** MFM can identify the true two causal variant model at smaller sample sizes than independent analysis in simulated data when there is sharing between diseases. **c,d** When there is no sharing (c) or one disease has no true associations (d), no information is gained by using MFM but there is only minimal loss in accuracy in doing so."*

For panel b, Disease 2 was on the left and Disease 1 on the right, though the simulation settings were listed correctly. We have now kept Disease 1 on the left for all panels.

9. Page 28 line 503: "The SNPs belonging to the above-mentioned groups" - why not use all 345 SNPs? Is that too computationally intensive? But then that makes the simulation not particularly realistic...

We have clarified this in the 2nd paragraph of "Simulations - single trait" in the Methods section:

"This extraction was done for computational efficiency, and is based on the previous analysis of MS and T1D that identified these SNP groups as contributing the majority of the posterior probability. All other SNPs contribute negligible posterior probability and we assume this in the simulations. The total number of SNPs in the region is not disregarded and is used in the prior probability calculations for the SNPs that are analysed. "

10. Page 29 lines 518-521. Does this mean you set one variant (the same variant) in a group as causal for both diseases? What is the rationale for doing this rather than keeping the different causal SNPs (within the same group) chosen for the different diseases?

Yes, when there was a shared causal variant, we set it to be the same for both diseases and add a rationale for this to the end of the 1st paragraph in "Simulations - multiple traits" of the Methods section:

"This reflects our belief that if causal variants for two diseases are known to belong to the same small SNP group, it is likely that the same SNP is causal for both diseases rather than different SNPs in the same high-LD group."

11. Page 38-40 (Tables 1 and 2) - while appreciating the rationale for comparing the "top" models from stepwise and stochastic search, or from stochastic search and MFM, actually this seems a bit of a simplistic comparison. One of the attractive features of stochastic search/MFM is that one does not have to choose the "top" model - one rather ends up with a set of competing models, each with their posterior probabilities.

Indeed, and we do give much more complete posterior distributions in the supplementary information, as well more complete summaries in a browsable form at <https://chr1swallace.github.io/MFM-output/>. However, for the purposes of summarising results in a space efficient format for a printed manuscript, we settled on top models for simplicity. We added the following text to the third paragraph of the Results section to highlight that this was a practical rather than ideal choice.

"While one of the strengths of Bayesian methods is that multiple competing models can be identified with posterior support for each, for the purposes of comparing stochastic search and stepwise search results, we chose to focus on discrepancies between the "best" models chosen for each."

Minor comments:

Page 34 line 605-606: "using the methods proposed in this paper" - I think you need to refer people to the Supplementary Note here, since that would appear to be the place where the MFM method is actually described.

We now point to the Supplementary note in paragraph 3 of "Fine mapping analyses of ImmunoChip-genotyped diseases":

"...using the methods proposed in this paper (details in Supplementary Note), as implemented in the R package MFM..."

Page 40-42 (Tables 2-4): There seems to be a disconnect between the figure legends and the table column headings. In the figure legends you use the descriptors "stochastic search" and "MFM" while in the Table column headings you seem to be using the terms "Independent" and "Joint". It would be helpful to use the same terms in the Table column headings as are used in the legends.

We have changed the legends and headings to ensure that the tables and figures use the same terms. In particular, Figure 4 and Tables 2-4 consistently use the terms "Independent" and "MFM".

Supplementary Note Page 1: you seem to be missing several members of the authorship?!

We have copied the author list on the manuscript to the Supplementary Note.

Supplementary Note Page 10: "have been derived [?]" - missing reference?

Thank you for noticing this LaTeX error and the missing references are now visible.

Legend to Suppl Fig 3: "MMPI > 0.5". (i.e. use > as opposed to the strange upside down ? sign).

Thank you for noticing this LaTeX error, and it has been corrected.

Reviewers' Comments:

Reviewer #1:

Remarks to the Author:

The authors have fully addressed my prior concerns, which were mainly requests for clarification. The authors' replies to my questions (and those of the other reviewer), and the revisions to the manuscript, were helpful. The revised manuscript is improved with the revisions.

Reviewer #2:

Remarks to the Author:

The authors have satisfactorily addressed all my previous comments.